Research

evolution, genomics

recombination suppression, Alauda, conservation, Cape Verde, island endemic, demography

**Author for correspondence:**
Elisa G. Dierickx
e-mail: elisa.dierickx@gmail.com

# Genetic diversity, demographic history and neo-sex chromosomes in the Critically Endangered Raso lark

Elisa G. Dierickx[1,2], Simon Yung Wa Sin[3,5], H. Pieter J. van Veelen[6,7], M. de L. Brooke[1], Yang Liu[4], Scott V. Edwards[3] and Simon H. Martin[1,8]

[1]Department of Zoology, University of Cambridge, Cambridge, UK
[2]Fauna and Flora International, Cambridge, UK
[3]Department of Organismic and Evolutionary Biology and Museum of Comparative Zoology, Harvard University, Cambridge, MA, USA
[4]Department of Ecology, Sun Yat-sen University, Guangzhou, People's Republic of China
[5]School of Biological Sciences, University of Hong Kong, Hong Kong, People's Republic of China
[6]Wetsus, European Centre of Excellence for Sustainable Water Technology, Leeuwarden, The Netherlands
[7]Groningen Institute for Evolutionary Life Sciences, University of Groningen, Groningen, The Netherlands
[8]Institute of Evolutionary Biology, University of Edinburgh, Edinburgh, UK

EGD, 0000-0002-4124-586X; HPJvV, 0000-0002-5766-599X; YL, 0000-0003-4580-5518; SVE, 0000-0003-2535-6217

Small effective population sizes could expose island species to inbreeding and loss of genetic variation. Here, we investigate factors shaping genetic diversity in the Raso lark, which has been restricted to a single islet for approximately 500 years, with a population size of a few hundred. We assembled a reference genome for the related Eurasian skylark and then assessed diversity and demographic history using RAD-seq data (75 samples from Raso larks and two related mainland species). We first identify broad tracts of suppressed recombination in females, indicating enlarged neo-sex chromosomes. We then show that genetic diversity across autosomes in the Raso lark is lower than in its mainland relatives, but inconsistent with long-term persistence at its current population size. Finally, we find that genetic signatures of the recent population contraction are overshadowed by an ancient expansion and persistence of a very large population until the human settlement of Cape Verde. Our findings show how genome-wide approaches to study endangered species can help avoid confounding effects of genome architecture on diversity estimates, and how present-day diversity can be shaped by ancient demographic events.

## 1. Introduction

Island species have suffered 89% of all recorded avian extinctions, despite only representing 20% of all bird species [1–3]. This reflects multiple factors, including vulnerability to alien invasive species [3] and intrinsic geographical characteristics of islands such as isolation and small distributional area, which can increase vulnerability to climate change [4] and limit dispersal into an alternative habitat. Species with a small effective population size ($N_e$) are also subject to three types of genetic risk: inbreeding depression through the exposure of deleterious recessive alleles and loss of heterozygote advantage [5,6]; accumulation of deleterious alleles due to increased drift (mutational meltdown) [7,8]; and loss of potentially adaptive genetic variation limiting future adaptive potential [9,10]. Previous studies have found that some, but not all, island species show reduced genetic diversity and increased inbreeding compared to their mainland counterparts [11,12]. However, this pattern may be driven more by the bottleneck associated with colonization rather than

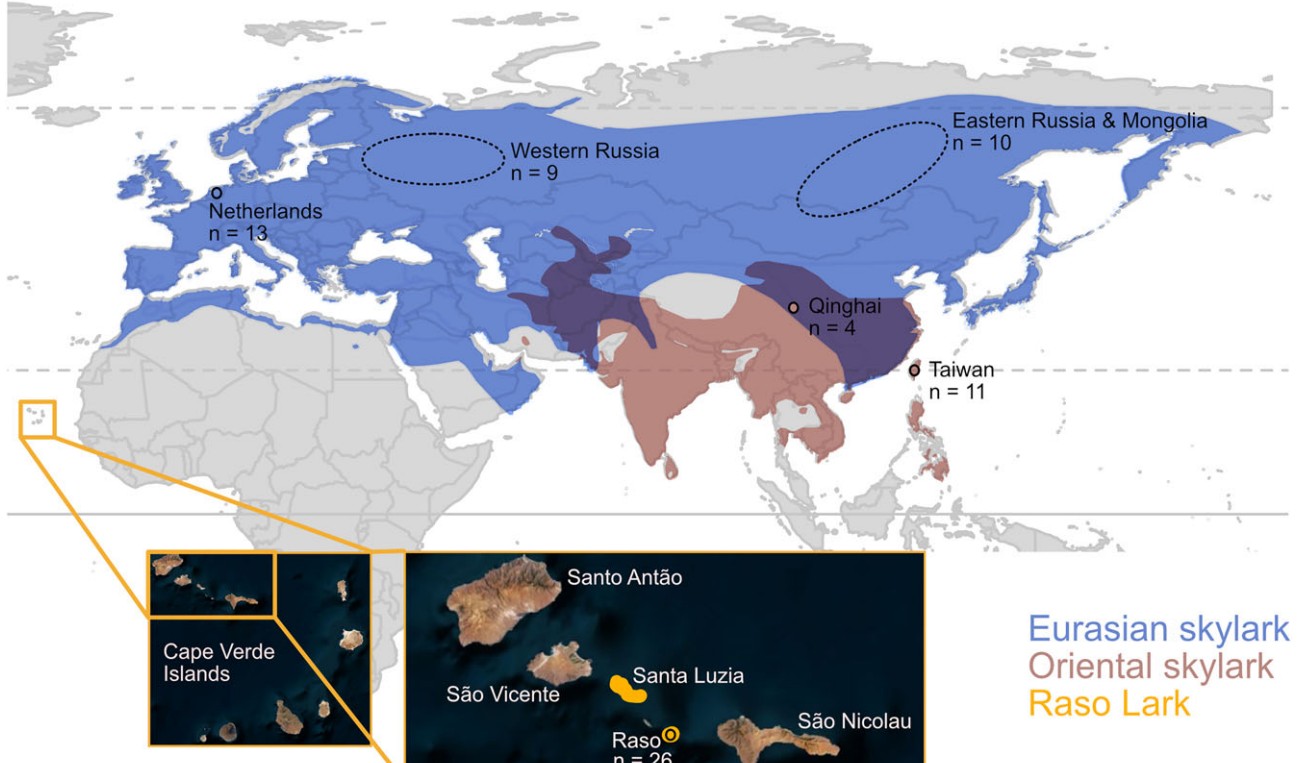

**Figure 1.** Species ranges, sampling locations and sample sizes. The insert shows the location of Raso in Cape Verde. Sampling areas for each of the six *Alauda* populations are indicated. See electronic supplementary material, table S2 for sample details. (Online version in colour.)

long-term reduction in $N_e$ [12]. For most species, poor census data makes it difficult to assess whether island existence *per se* is likely to expose species to the genetic risks above.

The Raso lark *Alauda razae* is endemic to the uninhabited 7 km$^2$ islet of Raso in Cape Verde [13]. Irregular counts since 1965 and yearly counts since 2002 indicate a small population fluctuating between approximately 50 and 1500 individuals [14–16] (electronic supplementary material, table S1). Sub-fossils indicate a larger past distribution encompassing neighbouring islands: Santa Luzia (35 km$^2$), São Vicente (227 km$^2$) and Santo Antão (779 km$^2$) [13] (figure 1). Raso larks disappeared abruptly from the neighbouring islands following the arrival of humans along with cats, dogs and rodents in 1462 [13]. Today, Raso is the last of the larger islets of Cape Verde that remains free of these mammals.

The genus *Alauda* has enlarged neo-sex chromosomes which appear to derive from ancestral autosomes [17–19]. Cessation of recombination should ultimately lead to sequence divergence between the neo-Z and neo-W chromosomes, and degeneration of the latter. However, in the short-term, before significant divergence has accumulated, suppressed recombination could effectively cause retention of distinct pre-existing alleles at homologous loci on the neo-W and neo-Z. This would also artificially elevate estimates of heterozygosity in females if the analysed loci are treated as diploid autosomes rather than two haploid gametologues. Indeed, Brooke *et al.* [19] observed microsatellite loci that would ordinarily be autosomal showing sex-linked segregation and excessive heterozygosity in female Raso larks. The authors hypothesized that the enlarged neo-sex chromosomes might therefore allow the retention of genetic variability in the face of population contraction. If the neo-sex chromosomes are indeed very large (as suggested by cytogenetic analysis; [18])

and if recombination is suppressed across much of their length, they could impact estimates of genetic diversity in Raso larks.

We therefore set out to investigate the contrasting effects of population contraction, which should reduce genetic diversity, and that of neo-sex chromosomes, which might preserve allelic variation, in the Raso lark. We produced a high-quality draft genome assembly for the related Eurasian skylark *Alauda arvensis* and used restriction site-associated DNA sequencing (RAD-seq) of 75 individuals from four lark species: Raso lark, Eurasian skylark, Oriental skylark *A. gulgula* and crested lark *Galerida cristata*. Our findings reveal the risk of confounding effects of neo-sex chromosomes on commonly used measures of diversity, as well as unexpected demographic events, and therefore highlight the value of genome-wide approaches to study diversity and genetic risks in endangered species.

## 2. Material and methods

All custom scripts and commands for all computational analyses are provided at https://github.com/simonhmartin/Raso_lark_diversity.

### (a) Sample collection

Twenty-six Raso lark blood samples were collected on Raso between 2002 and 2014. From colleagues, we also obtained blood and tissue samples for related species: 29 Eurasian skylarks, that we group into three populations, 15 Oriental skylarks from two locations, and 5 crested larks from Saudi Arabia (figure 1; electronic supplementary material, table S2). None of the sampled birds was likely to be a migrant based on the sampling dates (electronic supplementary material, table S2) and/or the migration pattern of the species. For samples

with unknown sex, sex was determined using PCR [20] and/or by examination of heterozygosity on the Z-linked scaffolds (see Results).

## (b) Whole-genome sequencing and assembly

A draft reference genome was obtained through whole-genome sequencing of a male Eurasian skylark (individual 0) collected in Mongolia (electronic supplementary material, table S2). Whole genomic DNA was isolated using DNeasy Blood & Tissue Kit (Qiagen, Venlo, The Netherlands) following the manufacturer's protocol. Two libraries were prepared: a 220 bp insert size fragment library using a PrepX ILM 32i DNA Library Kit for an Apollo 324 robot, following the manufacturer's protocol (TaKaRa, Kusatsu, Japan), and a 3 kb mate-pair library using an Illumina Nextera Mate Pair Sample Preparation kit and following the manufacturer's protocol (Illumina, San Diego, CA, USA). Both libraries were sequenced on an Illumina HiSeq 2500, producing 125 bp paired-end reads. Adaptor trimming and quality assessment were performed using Trimmomatic 0.32 [21] and FastQC [22], respectively, and Allpaths-LG [23] was used for assembly with option '–HAPLOIDIFY = TRUE'. Summary statistics were calculated with Allpaths-LG.

## (c) RAD library preparation

DNA for RAD-seq was extracted as above. Single digest RAD-seq libraries were prepared using *PstI*. Each individual was assigned an 8 bp inline barcode, and equimolar concentrations of 16 uniquely barcoded individuals were pooled and double-indexed by 16 cycles of high-fidelity PCR using Phusion High-Fidelity PCR Master Mix (Thermo Fisher Scientific, Waltham, MA, USA) with Illumina barcodes. The PCR products were pooled in equimolar quantities and sequenced on an Illumina HiSeq 1500, producing 100 bp single-end reads.

## (d) Sequence processing and alignment

We used *process_radtags* in Stacks 1.35 [24] without quality filters to sort sequence reads by barcode. We then used Trimmomatic to trim restriction sites (6 bp) and remove all trimmed reads shorter than 95 bp. *Process_radtags* was then used again to filter for quality. Reads were aligned to the Eurasian skylark genome using Bowtie 2 [25], and reads with multiple significant hits were removed.

## (e) Pseudo-chromosomal assembly

We inferred the approximate chromosomal location and orientation of each Eurasian skylark scaffold larger than 1 megabase (Mb) based on homology with the zebra finch *Taeniopygia guttata* genome [26] (i.e. a pseudo-chromosomal assembly). We used the *nucmer* tool in MUMmer v. 3.23 [27] to identify putative homologous regions. Alignments shorter than 5 kb were discarded using the *delta-filter* tool. We used *mummerplot* to visualize all alignments and determine the optimal scaffold order and orientation. Additional manual changes were made based on visual inspection of the scaffold arrangement.

## (f) Identification of sex-linked regions based on heterozygous genotypes

To identify putative sex-linked regions, we examined how the proportion of heterozygous sites varies across the genome in each individual. We called genotypes using GATK v. 3.4 [28,29] HaplotypeCaller (in GVCF mode) and GenotypeGVCFs, with default parameters. Thirteen individuals with poor coverage (less than 3 million sites with greater than or equal to 5× coverage, electronic supplementary material, table S2) were

excluded. The proportion of heterozygous genotype calls for each individual was computed for 250 kb windows across each scaffold considering only sites with greater than 5× coverage using the Python script popgenWindows.py (github.com/simonhmartin/genomics_general). Windows with fewer than 100 sites (with greater than or equal to 5× coverage) genotyped in the population (both variant and invariant) were excluded.

## (g) Relatedness

We estimated relatedness among individuals using two methods suited to low-coverage genomic data. The first was NgsRelate [30], which considers genotype likelihoods. These were computed using GATK as described above. NgsRelate was run considering sites covered by at least one read in each individual in the population. The second approach was KGD [31], which is designed for GBS data such as RAD-seq data, and also accounts for low sequencing depth. The input file was generated using ANGSD [32]. Only sites covered by at least 100 reads across the 26 Raso larks or 50 reads across the 10 Eurasian skylarks from the Netherlands were included. Following [31], we explored filtering options to find SNP subsets that gave realistic values of self-relatedness (approx. 1). The chosen filter was to use only SNPs with a Hardy–Weinberg disequilibrium value between 0 and 0.1.

## (h) Allele frequency spectra and genetic diversity

We used a two-step pipeline in ANGSD [32] to infer allele frequency spectra from the RAD-seq reads mapped to the reference genome, accounting for uncertainty in low-coverage sequencing data. Genotype likelihoods were inferred using ANGSD with likelihood method 2, and only sites with a base alignment quality (baq) greater than or equal to 1 and SNP quality greater than or equal to 20 were considered. A mapping quality adjustment was applied for reads with multiple mismatches (-C 50), following the author's recommendation. The *realSFS* tool was then used to infer the allele frequency spectrum with a maximum of 100 iterations, with 20 bootstraps. Nucleotide diversity ($\pi$) was computed from the frequency spectrum as the sum of the weighted products of the major and minor allele counts for each allele count category, including the zero category (invariant sites).

## (i) Demographic inference

We applied two different approaches to investigate historical demographic changes in the Raso lark based on the frequency spectrum (averaged across 20 bootstrap replicates). First, we used $\delta a \delta i$ [33] to compare four different models of increasing complexity. The first model imposes a constant population size. Since $\delta a \delta i$ only optimizes the shape of the frequency spectrum, this model has no free parameters. The second model adds a single change in population size at some point in the past (two free parameters: time and relative size of the new population). The third and fourth models each added an additional change (along with two free parameters). Model optimization was performed using grid sizes of 50, 60 and 70, and repeated 10–50 times to confirm optimization.

In the second approach, we used Stairway Plot [34] to infer the optimal population size history given the SFS. We used the 'two-epoch' model, with the recommended 67% of sites for training and 200 bootstraps. We tested four different numbers of random breakpoints: 12, 25, 37 and 50.

To convert inferred population sizes and times to numbers of individuals and years, respectively, we used the collared flycatcher *Ficedula albicollis* mutation rate estimate of $4.6 \times 10^{-9}$ per site per generation [35]. We estimated the generation time of the Raso lark, defined as the mean age of the parents of the

current cohort at age at first breeding + (1/mean annual mortality) [36]. This gave a generation time of 6.5 years.

## 3. Results

### (a) Enlarged neo-sex chromosomes in *Alauda* larks

We generated a high-quality draft genome assembly for a male Eurasian skylark based on 154 342 128 paired-end reads and 263 949 984 mate paired reads. The resulting 5714 scaffolds total 1.06 Gb with a scaffold N50 length of 1.44 Mb (71.5 Kb for contigs) and BUSCO [37] completeness score for the Aves gene set of 93%. We were able to arrange 311 scaffolds of greater than or equal to 1 Mb in length, totalling 648 Mb (63% of the genome), into pseudo-chromosomes based on homology with the zebrafinch genome (electronic supplementary material, figure S1).

RAD-seq reads for 75 individuals were mapped to the reference genome, yielding a high density of RAD loci sequenced to low depth (min = 1.5×, max = 6.9×, mean = 2.8×) (electronic supplementary material, table S2). The average proportion of reads mapped was 88% for Eurasian and Oriental skylarks, 89% for Raso larks and 82% for the outgroup crested larks. Using an arbitrary threshold of at least 100 reads across the dataset to designate a shared RAD locus yields 62 million genotyped sites across the dataset, of which 4 million are SNPs.

To investigate the composition of the neo-sex chromosomes, we examined the density of heterozygous genotypes across the genome in each individual. As females carry a single copy of the Z, they should have no heterozygous sites on this chromosome (with the exception of repetitive elements at which mis-mapping can occur, and possibly pseudo-autosomal regions where the W and Z retain homology). The same should be true for neo-sex chromosomes, unless recombination suppression is recent or incomplete, such that the gametologues retain a high level of sequence similarity. This would allow RAD-seq reads from the neo-W to map to scaffolds representing the neo-Z (there should be no neo-W scaffolds in our male reference genome). This could potentially lead to an elevation (rather than a reduction) in the density of heterozygous genotype calls in females. Indeed, we observe strong elevations in the density of heterozygous genotypes across large portions of chromosomes 3, 4a and 5 in all female Raso larks and not in males (figure 2a). This is consistent with the formation of neo-sex chromosomes and subsequent recombination suppression involving these three autosomes. The gametologues have evidently not diverged to the point that sequence identity has decreased significantly. In fact, they retain around 99% similarity (1.15 differences per 100 bp on average). It is important to note that the chromosome map used here—based on the zebra finch genome—does not reflect the true karyotype for larks. While previous work indicates that only a fragment of chromosome 4a has become sex-linked [17], we cannot currently say whether chromosomes 3 and 5 have undergone similar fragmentation. One scaffold appeared to bridge between regions of suppressed and normal recombination (electronic supplementary material, figure S1), but the two parts mapped to different chromosomes (3 and 2), so we concluded that this is most likely a misassembly. We also lack any direct evidence that chromosomes 3, 4a and 5 (or fragments thereof) have fused to the Z chromosome. One scaffold appears to bridge chromosomes 3 and 5, but this too may be a misassembly, as it maps to the centre of the two regions of suppressed recombination (electronic supplementary material, figure S1).

Although we lack information on how chromosome structure has evolved, comparison with the other species (figure 2a; electronic supplementary material, figure S2), gives insights into the progression of recombination suppression through time. An increased density of heterozygous genotypes in females exists across the same three chromosomes in Eurasian skylarks from the Netherlands, though it is not as easily distinguishable from the higher genome-wide background heterozygosity in this population (figure 2a). Within the regions of suppressed recombination, the density of heterozygous sites is similar to that in Raso lark females (1.09%). Female Oriental skylarks and crested larks also show elevated densities of heterozygous sites, but over smaller portions of chromosomes 3 and 5 (figure 2a). Using our results in combination with those from a recent study of two additional outgroup species, the bearded reedling *Panurus biarmicus* and the horned lark *Eremophila alpestris* [38], we are able to partially reconstruct the stepwise progression of recombination suppression (figure 2b). We infer that the initial suppression occurred in two narrow 'strata' on chromosomes 3 and 4a-1 [38]. This increased marginally on chr. 3 in the common ancestor of the larks, and was followed by the formation of a large new stratum across most of chr. 5 in the common ancestor of the crested lark and the *Alauda* larks. Further recombination suppression in several parts of chr. 3 then probably occurred in the ancestor of *Alauda*. However, this was probably polymorphic, because it is lacking in the Oriental skylark, and also in the eastern populations of the Eurasian skylark, but it is present in the western populations of the Eurasian skylark as well as in the Raso lark (electronic supplementary material, figure S2). These species differences cannot easily be explained by artefacts such as allele dropout, as the trends we describe are at the scale of multiple megabases, and are therefore supported by tens to hundreds of separate RAD loci.

### (b) Genetic diversity

To study genetic diversity and demography, we used only the genomic scaffolds that show no evidence of suppressed recombination (electronic supplementary material, figure S3). We first screened for relatedness and found that most of the 26 Raso lark samples show little or no detectable relatedness, but three pairs of individuals have levels of relatedness of approximately 0.5, indicating either sibling or parent–offspring relationships, while several other pairs have non-zero relatedness of up to approximately 0.2 (electronic supplementary material, figure S4). We chose not to exclude these from the downstream diversity analyses as this level of relatedness probably reflects the true composition of the population rather than a sampling artefact. Indeed, one of the pairs of close relatives consists of individuals sampled 3 years apart, in 2011 and 2014. Sampling dates of the other pairs could not be verified due to a labelling error of some DNA samples. The equivalent analyses performed for the 13 Eurasian skylarks from the Netherlands found no consistent evidence of high relatedness (electronic supplementary material, figure S4).

Average nucleotide diversity (*π*) across the 26 Raso larks based on inferred allele frequency spectra is 0.001 (electronic

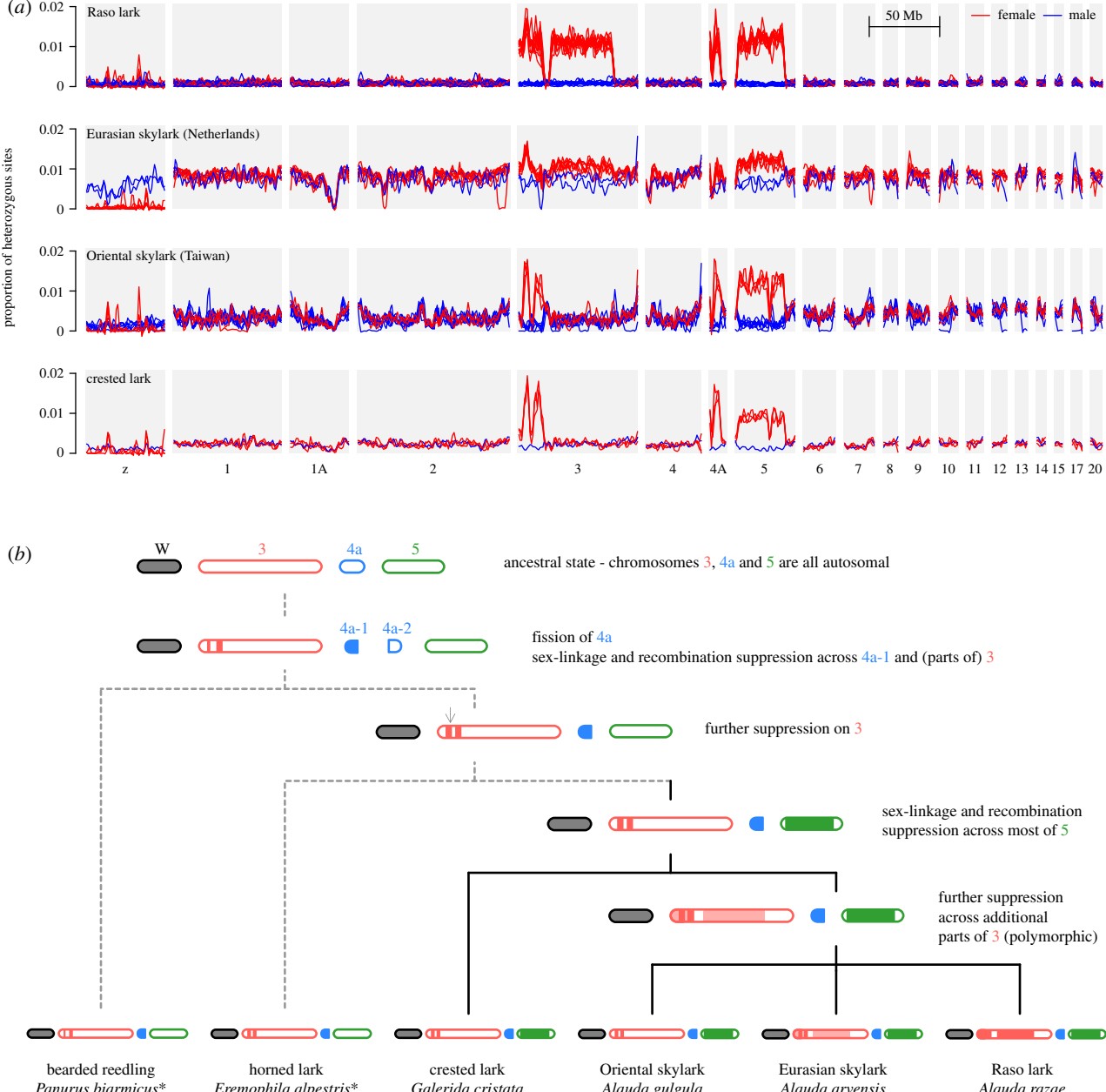

**Figure 2.** Individual heterozygosity reveals neo-sex chromosomes. (*a*) The proportion of heterozygous sites per 250 kb window in each individual, plotted across each pseudo-chromosome (assembled based on homology with the zebra finch karyotype), with locally weighted smoothing (loess, span = 10 Mb). Females and males are indicated by red and blue lines, respectively. One population from each species is shown (see electronic supplementary material, figure S2 for all populations). A lack of heterozygous genotypes in females on the ancestral Z chromosome indicates that it is haploid due to divergence and degeneration of the ancestral W (the narrow peaks most likely reflect collapsed repeats in our assembly). A high density of heterozygous genotypes in females on autosomes is indicative of more-recent recombination suppression between neo-W and neo-Z, without significant degradation of the neo-W. (*b*) Model for the progression of recombination suppression between lark neo-sex chromosomes. Shading indicates inferred neo-W regions with suppressed recombination in females based on our analysis (panel (*a*) and electronic supplementary material, figure S2) as well as reference [38] (the latter are shown in grey and indicated with an asterisk). Note that chromosomes are not shown as fused as we do not have evidence for or against physical linkage in this study. The fragmentation of chromosome 4a is based on previous linkage mapping [17]. Light shading indicates polymorphic recombination suppression (i.e. in some populations but not others; electronic supplementary material, figure S2). (Online version in colour.)

supplementary material, table S3). This is just less than 10% of that in Eurasian skylark from the Netherlands (0.0114), and 25% of that in the Oriental skylark from Taiwan (0.0041). Assuming an equilibrium population with $\Theta = 4N_e\mu$ and a per-generation mutation rate equivalent to that of the collared flycatcher, this translates to an effective population size ($N_e$) of approximately 50 000 in the Raso lark, compared to approximately 500 000 in the Eurasian skylark. Therefore, genetic diversity in the Raso lark, despite being one-tenth of that in the Eurasian skylark, appears to reflect

a far larger historical effective population size than its current census size of approximately 1000 individuals.

We also considered how different the estimated genetic diversity of Raso larks would be if we did not account for the enlarged neo-sex chromosomes (as may easily occur using a RAD-seq approach without a reference assembly). Although the regions of suppressed recombination represent 12% of the genome, estimated genetic diversity is nearly doubled (0.0019) in our dataset of 15 females and 11 males when these regions are not excluded.

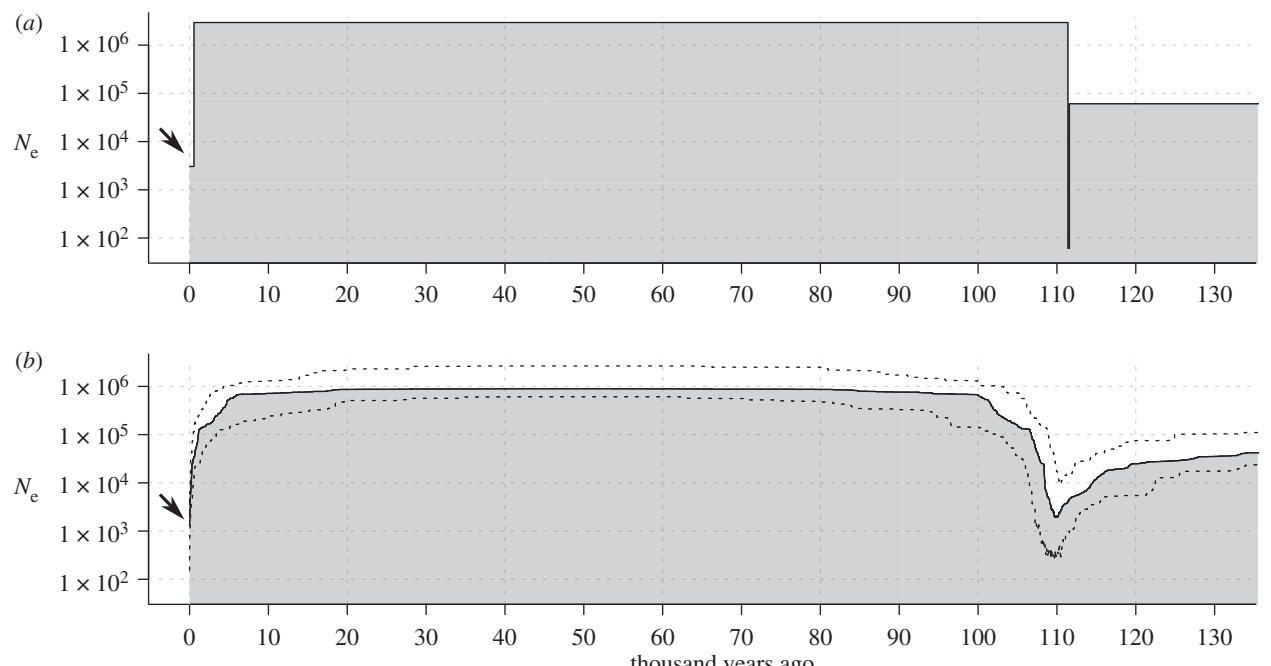

**Figure 3.** Demographic model fitting supports ancient and recent bottlenecks. The best fit models for inferred $N_e$ (log scale) over time (with the present on the left), based on inference using (*a*) $\delta a \delta i$ [33] and (*b*) Stairwayplot [34]. Dashed lines indicate the 95% confidence interval. Arrows indicate the recent population contraction inferred using both methods.

## (c) Genomic signatures of ancient and recent bottlenecks

Although autosomal genetic diversity in Raso larks is lower than that in related species, it remains far greater than would be predicted under long-term persistence at its small census size of typically under 1000. We therefore hypothesized that a population contraction from a larger ancestral size occurred recently enough that some of the pre-existing genetic diversity has been retained. To test this hypothesis, and investigate to what extent a recent population collapse has impacted Raso lark genetic variation, we examined the allele frequency spectrum, which carries information about historical population size changes [39]. Surprisingly, the frequency spectrum (computed using only scaffolds showing no evidence of suppressed recombination; see electronic supplementary material, figure S3) is skewed towards an excess of rare variants (electronic supplementary material, figure S5), which is also captured in the negative Tajima's *D* value of −0.69. An excess of rare variants is typical of population expansion rather than contraction and is therefore not consistent with our understanding of the recent history of the Raso lark. This skew was consistent across 20 bootstrap replicates and 26 'drop-one-out' replicates, in which a single individual was excluded in each case (electronic supplementary material, figure S5), which demonstrates that it is not a sampling artefact.

We therefore used two related approaches to explore the historical demography of this species based on the frequency spectrum. First, we used $\delta a \delta i$ [33] to compare the fit of simple models allowing zero, one, two or three changes in population size in the past. The simplest model imposes a constant population size, and shows a very poor fit to the data, as expected (electronic supplementary material, figure S6). A model that allows a single change in population size in the past shows a far better fit and a greatly improved composite likelihood

(electronic supplementary material, figure S6). The inferred model involves an ancient population expansion rather than a recent contraction (electronic supplementary material, figure S6). This one-change model is able to recreate the excess of singleton variants in Raso larks, but still shows a fairly poor fit for to the frequency of other rare variants. The model allowing two changes in population size again shows a major increase in the likelihood and better fit to the frequency spectrum (electronic supplementary material, figure S6). However, again the inferred demographic model does not include a recent contraction. Instead, it consists of an ancient bottleneck followed by an expansion approximately 110 000 years ago. These findings show that the distribution of genetic variation in the Raso lark can be explained fairly precisely by a few ancient demographic changes, without the inclusion of a recent population contraction.

Given the recent census estimates of approximately 1000 individuals for the Raso lark, we attempted to fit a final model that allows for a recent population contraction. As $\delta a \delta i$ failed to optimize a model with a third change in population size (i.e. two additional free parameters), we instead used a fixed population contraction, and performed a manual search to approximate this parameter estimate by re-running the optimization across a range of final $N_e$ values from 0.1% to 1% of the ancestral size. The resulting best fit model once again shows an improved fit to the data over models without a recent contraction, although it results in minimal detectable difference to the expected frequency spectrum (electronic supplementary material, figure S6). The model again involves an ancestral bottleneck that ended over 100 000 years ago, expanding to approximately 3 million individuals thereafter, with a recent contraction down to about 3000 individuals that occurred just 550 years (around 85 generations) ago (figure 3*a*). Although parameter estimates are not to be interpreted as exact, we note that this date matches nearly perfectly with the arrival of human

settlers in Cape Verde in 1462. In summary, these models show that the distribution of genetic variation in the Raso lark is consistent with a recent and dramatic population contraction, but that the overall distribution of genetic diversity in this species is more significantly shaped by expansion after an ancient bottleneck.

Our second approach used Stairway Plot [34] to estimate the optimal population size history given the frequency spectrum. The inferred history is remarkably similar in its general structure to the 3-change model inferred using $\delta a \delta i$, with a strong ancestral bottleneck around 110 000 years ago followed by expansion to a large population of nearly a million individuals, with a sharp recent contraction (figure 3b). The contraction is most pronounced in the most recent past, but it is inferred to have initiated further in the past, approximately 5000 years ago. Given the $\delta a \delta i$ results showing that any signal of the most recent population contraction in the frequency spectrum is fairly weak, it is likely that the exact timing of this event would be difficult to infer. Nevertheless, both approaches indicate that the Raso lark population was probably very large up until fairly recently, thus agreeing with our second hypothesis that the population still retains much genetic variation that pre-dates its recent contraction.

## 4. Discussion

Genetic markers have long been used to investigate the genetic risks faced by species thought to have small $N_e$, such as island endemics [10–12]. Genomic approaches now allow us to address these questions at far greater resolution, revealing how different parts of the genome have been affected, and inferring past demography. We find that the island-endemic Raso lark has reduced genetic diversity compared to its widespread *Alauda* relatives, but that this difference is much smaller than the difference in census population sizes. Diversity in the Eurasian skylark is high for a passerine, being similar to that in the zebra finch (0.01 [40]) and double that in the house finch *Haemorhous mexicanus* (0.005 [41]). Diversity in the Raso lark is approximately one-tenth of that in the skylark, but is similar to that in the hihi *Notiomystis cincta*, another island-restricted species (0.00095 [10]). There are an estimated 1 million breeding pairs of Eurasian skylarks in the United Kingdom alone [42], and usually fewer than 1000 Raso larks on Raso. Our findings indicate that the Raso larks retain some ancestral diversity due to the recency of their population contraction from a much larger size. In addition, our study demonstrates the importance of understanding genome structure for studies of genetic diversity. The enlarged neo-sex chromosomes of Raso larks might have led to erroneous conclusions had they not been accounted for, but they are also intriguing as a potential source of functional allelic variation [19].

Previous work suggested that neo-W and neo-Z chromosomes arose at the base of the *Sylvioidea* about 40 Ma through the fusion of part of chromosome 4a to both the W and Z [17]. Our results indicate that two other large chromosomes, 3 and 5, have also become sex-linked in the genera *Alauda* and *Galerida*. Another study conducted in parallel [38] supports these observations and further indicates that at least chromosome 3 is also sex-linked in two additional outgroup species (horned lark, bearded reedling). Without a chromosomal assembly, we cannot currently determine whether homologs of chromosomes 3 and 5 have fused to the W and Z (as is

inferred to be the case for chromosome 4a-1; [17]), or whether they segregate as multiple sex chromosomes, as is seen in some other taxa [43,44], but the visually obvious enlargement of both the W and Z chromosomes in larks [18] is consistent with fusions.

Despite the deep age of the neo-sex chromosomes, our results indicate fairly high sequence similarity of approximately 99% between the neo-W and neo-Z in the regions of recombination suppression. This value may be somewhat underestimated, as the most divergent parts would be excluded due to poor read mapping [38]. Nevertheless, even a divergence of 2% would translate to approximately 2.2 million generations to coalescence—much less than the age of the neo-sex chromosomes. Importantly, this coalescence time reflects not the age of the fusions but rather the age and extent of recombination suppression. Recombination and gene conversion may continue to occur at low levels even between divergent parts of the gametologues. Moreover, the differences in the extent of recombination suppression among species indicates that it has accumulated in a stepwise manner through the formation of 'strata'. Indeed, we even find evidence for variation in the extent of recombination among populations of Eurasian skylarks, implying that recombination suppression is not an instantaneous process.

The evolutionary causes and consequences of the enlarged neo-sex chromosomes require further study. It is hypothesized that sexually antagonistic selection might favour recombination suppression between sex chromosomes [45]. Whether this is true in the *Alauda* larks is currently unclear. It was previously hypothesized that—irrespective of the original cause of recombination suppression—the maintenance of pre-existing allelic variation between the gametologues might now contribute to functional diversity and therefore fitness in female Raso larks [19]. This is analogous to the retention of heterozygosity in functionally asexual hybrid plants [46], except that here it applies only to a portion of the genome, and only in females. While a non-recombining W chromosome is expected to eventually degenerate through gene loss and accumulation of deleterious mutations, the retention of nearly 99% sequence identity between the non-recombining portions of neo-W and neo-Z means that we cannot rule out the presence of functional female-specific alleles on the neo-W. Future work should test this hypothesis by investigating whether homologous genes on both gametologues are still expressed and contribute to female fitness.

When we consider only the autosomes, we find that levels of diversity in Raso larks are higher than would be predicted with long-term persistence at their current population size. Such a mismatch between differences in genetic diversity and census population size is a well-documented phenomenon [47]. Research in this field is typically aimed at explaining lower-than-expected diversity in larger populations, either through a mismatch between census and effective population sizes [48], or through increased action of selection affecting linked sites [49]. Higher-than-expected diversity in smaller populations requires a different explanation, such as gene flow from other populations, increased mutation rate or recent contraction from a larger ancestral size [12]. In the case of the Raso lark, gene flow is unlikely, given that the other two *Alauda* species are largely restricted to Eurasia. We cannot currently rule out an increase in mutation rate, but there is strong evidence from our demographic modelling that the population recently contracted from a much larger

size, making this the most parsimonious explanation. Assuming this contraction coincided with the settlement of Cape Verde in 1462, the fairly long average generation time of Raso larks of 6.5 years means that this would correspond to just 85 generations ago. Under a simple model of loss of $1/2N$ times the ancestral variation through drift each generation, with a current $N_e$ of 1000, this translates to a retention of over 95% of the pre-existing genetic diversity (electronic supplementary material, figure S7). Even in the more extreme case of $N_e = 100$, more than 60% of the diversity is retained.

While it may be unsurprising that the recent contraction has failed to eliminate genetic diversity, we were surprised by the skew towards rare variants in the allele frequency spectrum, indicating that the genetic make-up of this population is largely shaped by a major population expansion that occurred deeper in the past (estimated at approx. 110 000 years ago). One possibility is that this could coincide with the larks' colonization of Cape Verde, but it is likely that divergence from the Eurasian skylark occurred much earlier. This question is the focus of an ongoing study. Summing the areas of the surrounding islands, Santa Luzia, São Vicente and Santo Antão [13], we can predict a minimum ancestral range of 1048 km², which is approximately 150 times the area of Raso (although this rough estimate does not consider habitat suitability). Extrapolating from the current population of ≈1000 predicts an ancestral population size of 150 000. Our modelling estimates a much larger $N_e$ of nearly a million prior to the recent collapse. It is possible that Raso larks were more abundant due to higher population density in the past, or that their colonization of Cape Verde from a larger mainland population occurred more recently.

Despite the considerable genetic diversity in Raso larks relative to their population size, continued existence at this size will inevitably increase their genetic risks. We found three pairs of closely related Raso larks out of 26 sampled. Although we cannot rule out a chance effect, finding related individuals is unsurprising given that the population dropped to just 57 individuals in 2004, around two generations ago. With sufficient sample sizes, individual-level relatedness may provide more sensitive detection of recent population collapse than population-level diversity, which can take multiple generations to change appreciably. The remaining genetic diversity in Raso Larks is likely to be of critical importance for their future adaptive potential [10].

This is especially relevant given the vulnerability of Cape Verde to environmental changes such as climate change [50] and the ongoing reintroduction of Raso larks to the nearby island of Santa Luzia, initiated in April 2018.

Ethics. The Raso lark blood samples were obtained by a licensed bird ringer (M.d.L.B.: British Trust for Ornithology permit A 1871 MP) without damaging the health of the birds and with permission from the Cape Verdean authorities (Direção Nacional do Ambiente). The Oriental skylark blood samples were obtained without damaging the health of the birds and with permission from the Forest Bureau of Qinghai Haibei Autonomous Prefecture. Eurasian skylark samples from The Netherlands were collected under animal welfare license DEC6619B/C of the Institutional Animal Care and Use Committee of the University of Groningen, obeying Dutch Law. Permission to work with wild birds in Saudi Arabia was obtained from the National Wildlife Research Centre.

Data accessibility. All raw sequencing reads and the Eurasian skylark genome will be made freely available via GenBank and the European Nucleotide Archive (ENA, project accession PRJEB36048). All custom scripts, as well as the commands used to generate the results, are available online at https://github.com/simonhmartin/Raso_lark_diversity. Processed data files and all commands used to generate them are available on the Dryad Digital Repository: https://doi.org/10.5061/dryad.rg576jq [51].

Authors' contributions. E.G.D. designed the study and did the RAD-seq laboratory work, S.H.M. and E.G.D. analysed the data, M.deL.B. and E.G.D. collected the Raso lark samples, S.Y.W.S. led on the Eurasian skylark genome sequencing and assembly, S.V.E. and S.Y.W.S. advised on the RAD-seq and Eurasian skylark genome work, S.V.E. provided laboratory and computing resources, H.P.J.v.V. provided the Eurasian skylark samples from the Netherlands, Y.L. provided the other Eurasian skylark and Oriental skylark samples, E.G.D. and S.H.M. wrote the paper, and all authors commented on the manuscript.

Competing interests. We declare we have no competing interests.

Funding. This work has been generously supported by the Sir Peter Scott Studentship and the Rouse Ball Eddington Fund of Trinity College, Cambridge (to E.G.D.), the VOCATIO Award (to E.G.D.), the William Bateson Fellowship of St John's College, Cambridge (to S.H.M.), Julian Francis, RSPB, CEPF and BirdLife International's Preventing Extinctions Initiative.

Acknowledgements. Nick Horrocks and Per Alström shared crested lark and Eurasian skylark samples. Allison Shultz provided feedback over the course of this project. Marco van der Velde and Sarah Barker advised on genetic sexing and on RAD library preparation. Chris Jiggins provided use of his laboratory and computational resources. Paul Sunnucks and Roberta Bergero provided helpful comments. Thanks to the field assistants on Raso: Mark Bolton, Ewan Campbell, Simon Davies, Mike Finnie, Tom Flower, Lee Gregory, Sabine Hille, Mark Mainwaring, Jason Moss and Justin Welbergen.

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
