## [Reviewer comments · Proceedings of the Royal Society B: Biological Sciences]

Review History

RSPB-2019-1102.R0 (Original submission)

Review form: Reviewer 1

Recommendation

Major revision is needed (please make suggestions in comments)

Scientific importance: Is the manuscript an original and important contribution to its field?

Acceptable

General interest: Is the paper of sufficient general interest?

Good

Quality of the paper: Is the overall quality of the paper suitable?

Marginal

Is the length of the paper justified?

Yes

Should the paper be seen by a specialist statistical reviewer?

No

Do you have any concerns about statistical analyses in this paper? If so, please specify them explicitly in your report.

No

It is a condition of publication that authors make their supporting data, code and materials available - either as supplementary material or hosted in an external repository. Please rate, if applicable, the supporting data on the following criteria.

Is it accessible?

N/A

Is it clear?

N/A

Is it adequate?

Yes

Do you have any ethical concerns with this paper?

No

Comments to the Author

The manuscript "Neo-sex chromosomes and demography shape genetic diversity in the Critically Endangered Raso Lark" by Dierickx et al. aimed at understanding the determinants of genetic diversity in the small island population of Raso lark by specifically looking at the role of neo-sex chromosomes and demography.

The authors detect regions of the genome as neo-sex chromosomes using the higher observed heterozygosity in female individuals. They report genetic diversity for the Raso lark and argue that the observed diversity is higher than what would be expected for an island species of such a small population size. They go on to show that by excluding the neo-sex chromosomes, genetic diversity is halved in the Raso lark and consider neo-sex chromosomes as a source of genetic variation. To further explain the observed level of genetic diversity, they conduct demographic modeling in which they show that demographic history of the species is best explained by incorporating both population contraction and an ancient expansion leading to an increase in rare variants.

The authors produced a draft genome assembly for the Eurasian skylark and RAD-seq reads for 78 individuals of 4 lark species which can be used as a valuable resource for further studies on larks and other avian comparative studies.

However, my main concern regarding this manuscript is that the elevated heterozygosity reported in Raso larks is obtained by calculating heterozygosity in neo-sex chromosomes in females. I think this is not correct since the neo-Z and neo-W chromosomes are diverged from one another and this means that the divergence between neo-Z and neo-W chromosomes contribute to the heterozygosity, i.e., $dx_y = 4N_{eu} + 2ut$ which leads to an overestimation of the effective population size. Moreover, authors argue that recombination suppression has retained genetic variation while this is a known fact that recombination suppression leads to the reduction in genetic diversity. Since the role of neo-sex chromosomes in maintaining genetic variation is a core part of the paper, I am afraid that this result has been incorrectly obtained by using the divergence between neo-Z and neo-W in females.

I think this manuscript is an interesting work for the Proceedings of the Royal Society B, however, it requires major revisions before it can be considered for publication as detailed in the comments below:

Introduction:

1. Line 55: Reference 12 gives a mixed message about the genetic diversity and N_e of island species. According to reference 12, some island species show reduced genetic diversity and some show no differences compared to mainland species. I think it would be more comprehensive if both aspects are mentioned in the introduction.
2. Line 70: A reference is required at the end of the sentence: "Previous work suggests that a change to genome architecture might buffer Raso larks from genetic diversity loss."
3. Lines 71: I think a schematic figure of the *Alauda* neo-sex chromosomes in the introduction would help the reader to follow the rest of the paper more easily. It would be helpful if the figure shows the recombining and non-recombining part of the neo-sex chromosomes together with the segment used for measuring genetic diversity in females.
4. Line 71 to 74: It is stated that "Cessation of recombination on neo-sex chromosomes could represent a source of heterozygosity, because females (the heterogametic sex) could retain distinct alleles at homologous loci on their neo-Z and neo-W chromosomes." I am confused about this statement. By homologous loci on neo-Z and neo-W chromosomes, are the authors referring to the so-called gametologous genes (homologous genes on the non-recombining sex chromosomes that are not yet degenerated from the W)? First, cessation of recombination leads to a decreased effective population size and therefore, reduction in heterozygosity. Second, due to the lack of recombination, the neo-Z and neo-W sequences have diverged from one another. All females are heterozygous not because two alleles are segregating in the population but because all W sequences are fixed for one allele and all Z sequences for the other allele. This is the result of divergence of two sequences since the cessation of recombination prior to the split of the species studied. It is not therefore correct to measure heterozygosity for females using neo-Z and neo-W sequences since the Tajim's π should tell us about the amount of genetic diversity since the most recent common ancestor of the species of interest. Please see for example the analyses in Bachtrog and Charlesworth 2000 regarding neo-sex chromosomes.

Methods:

5. Line 95: It is very good that all scripts are provided in the github page. A master script is however necessary which explains the order scripts should be run. For example, if a researcher downloads the raw data of this study from ENA and wants to reproduce the study, that master script should provide the steps in the pipeline.
6. I suggest a reordering of the paragraphs in the Methods section as follows:
 - Sample collection
 - Whole genome sequencing and assembly
 - RAD library preparation
 - Sequence processing and alignment
 - Pseudo-chromosomal assembly
 - Relatedness
 - "Allele frequency spectra and genetic diversity" merged with "Proportion of heterozygous sites across the genome" into one subheading
 - Demographic inference

Results:

7. Since the sequence of neo-sex chromosomes of several species is available here, it is possible to get an estimate of the timing of recombination suppression by calculating d_s . The authors could add this analysis to their manuscript to improve the inference of the timing of recombination suppression events.

8. Line 217: What is the percentage of mapped reads from Raso lark to the Eurasian Skylark assembly?

9. Line 223: Please report Tajima's π and Watterson θ for each case in males and separately for autosomes and neo-Z chromosomes.

10. Line 225: Genetic diversity in the Eurasian skylark from the Netherlands is 0.0097. This tells us about 1 heterozygous SNP in almost every 100 base pairs. This is really a high level of genetic diversity. What is the range of genetic diversity across other studied passerines? How does this high level of genetic diversity might have affected assembly construction? Did the high level of heterozygosity provide some challenges for the mapping of reads from Raso larks to the Eurasian Skylark assembly?

11. Line 226: Equation $\theta = 4Neu$ is used to obtain Ne from genetic diversity. The genetic diversity used is 0.0018 which contains the heterozygosity obtained from neo-sex chromosomes in females. I think this might not be correct since the Ne obtained from $\theta = 4Neu$ is the time to the most recent common ancestor of the sample. The diversity obtained from the neo-sex chromosomes in females is actually $dxy = 4Neu + 2ut$. This means that heterozygosity is overestimated by taking into account divergence between sequences leading to an overestimation of Ne . I think higher female heterozygosity must only be used as a method to detect suppressed recombination. Then to calculate heterozygosity in Raso lark, neo-sex chromosomes must be masked and only diversity of autosomes should be used. If sex chromosomes are to be used, then diversity should be calculated for the Z chromosome and W chromosome separately and in calculating Ne , their relative numbers for every male and female should be taken into account. For example, in a standard case, there are 3 Z and 1 W for every 4 autosomes for a male and a female in the population.

12. Line 245: The heterozygosity measured between neo-Z and neo-W chromosomes in females cannot be compared with heterozygosity between two neo-Z in males. The reason is that the heterozygosity obtained in females is actually divergence between the Z and W sequences. Once recombination is suppressed, the Z and W chromosomes start to differentiate from one another. This is similar to two subpopulations between which migration stops. To calculate genetic diversity, one should use only the Z sequences or the W sequences. Therefore, while elevated heterozygosity in females can be used as a technical way to find regions with suppressed recombination, it does not mean that we have maintenance of diversity due to recombination suppression.

13. Lines 249 to 258: From "Previously, only party of ..." are not the results of this paper or are the discussion regarding the result, I suggest to move this part to the discussion.

14. Lines 275 to 279: From "A similar analysis ..." to "... that excludes chromosomes 5 [38]" are not results of this paper and I suggest to move this section to the discussion.

15. Line 303: This π without the neo-sex chromosomes is the correct π and this should be reported as the heterozygosity in Raso Larks.

16. Line 304: "Recombination suppression across the neo-sex chromosomes therefore does indeed contribute to the maintenance of genetic variation in the Raso lark.". This sentence is not correct. Recombination suppression has led to a decrease in genetic variation, as mentioned before, one cannot use divergence between two sequences to calculate heterozygosity. I suggest to remove this sentence.

17. Line 312: It is stated that diversity difference between the Raso lark and the Eurasian skylark "is far greater than expected given the difference in current population sizes between the two species". This is not surprising because it is the Ne that matters not the census population size.

For example, census population size of humans is about 8 billion, that of *Pan troglodytes* is in the order of tens of thousands only, yet genetic diversity in chimps is higher than that of humans due to their higher N_e . What is “expected” as a relationship between the census size and genetic diversity?

18. Line 391: I suggest that the Relatedness section to be moved to the beginning of results as it is more of a quality check and it is important to be presented before the estimates of diversity.

Discussion:

19. Line 413: “... this difference is smaller than expected.”. What is the expected difference?

20. Line 436: “... 1-1.5% divergence”. This is a result not reported in the result section. Please add to both Methods and Results the text regarding the calculation of divergence between the neo-Z and neo-W chromosomes.

21. Line 451: “... has been favored by selection following the loss of genetic diversity.” Please clarify this sentence.

Line 471: “... level of divergence between the neo-Z and neo-W is similar in all three *Aluade* species.”. This is a result not presented in the result section. If known from previous work, please cite the reference.

Review form: Reviewer 2

Recommendation

Accept with minor revision (please list in comments)

Scientific importance: Is the manuscript an original and important contribution to its field?

Acceptable

General interest: Is the paper of sufficient general interest?

Acceptable

Quality of the paper: Is the overall quality of the paper suitable?

Good

Is the length of the paper justified?

Yes

Should the paper be seen by a specialist statistical reviewer?

No

Do you have any concerns about statistical analyses in this paper? If so, please specify them explicitly in your report.

No

It is a condition of publication that authors make their supporting data, code and materials available - either as supplementary material or hosted in an external repository. Please rate, if applicable, the supporting data on the following criteria.

Is it accessible?

Yes

Is it clear?

Yes

Is it adequate?

Yes

Do you have any ethical concerns with this paper?

No

Comments to the Author

Dear Editor:

Dierickx et al. profiled the genetic diversity of different lark population, particularly the endangered Raso Lark concentrated at two islands of Africa, regarding their chromosome-wide polymorphism level. They found signatures of neo-sex chromosomes, which are different between different skylark and lark populations. Then they tried different methods to fit the historical change of population size to the unexpectedly high polymorphism level as compared to its reported population size nowadays. Maybe because the population contraction is too recent, the current data cannot find a clear signature for the recent population contraction. I have several comments about the paper:

1. It is a bit confusing to state that neo-sex chromosomes maintain the heterozygosities in Raso lark, as the author mentioned in several places in the paper. As first neo-sex chromosomes probably formed by genetic drift in this case, due to the population size reduction. And generally, the differences between neo-Z and neo-W are expected to be excessive deleterious mutations that are fixed on the neo-W, rather than something that the population would like to maintain.
2. Given the current chromosomal sequences are using zebra finch as the reference (as the author also pointed out), it is very likely that the current fusion order in Figure 4 would be subjected to change. First, it is difficult to imagine that chr5, which is more distant from chrW than chr3, to first suppress recombination. Because one would expect sexual antagonistic selection would first accumulate nearby the W first, so that they are more likely to be inherited in only one sex. Second, there is a dip of heterozygosity in most populations between the first part and second part of chr3 (Figure 3, Raso lark). Is this caused by sequencing/assembly gaps? And also, is the female heterozygosity level of the first part of chr3 significantly different from those of the second part? This is important if they actually formed at separate time points. Another question, does the author have some explanation for the elevated heterozygosity level of male on the Z chromosome in Eurasian skylark? This is too obvious to be ignored.
3. Regarding the demographic analyses, I would think the most straightforward way is collect data from nearby islands that used to be inhabited by Raso lark? Would this be possible from the museum specimen?

Review form: Reviewer 3

Recommendation

Accept with minor revision (please list in comments)

Scientific importance: Is the manuscript an original and important contribution to its field?

Acceptable

General interest: Is the paper of sufficient general interest?

Good

Quality of the paper: Is the overall quality of the paper suitable?

Acceptable

Is the length of the paper justified?

Yes

Should the paper be seen by a specialist statistical reviewer?

No

Do you have any concerns about statistical analyses in this paper? If so, please specify them explicitly in your report.

No

It is a condition of publication that authors make their supporting data, code and materials available - either as supplementary material or hosted in an external repository. Please rate, if applicable, the supporting data on the following criteria.

Is it accessible?

Yes

Is it clear?

Yes

Is it adequate?

Yes

Do you have any ethical concerns with this paper?

No

Comments to the Author

Dierickx et al. used reduced representation (RAD) sequencing to explore genetic diversity and the population history of raso larks. They used the reference from a close relative, the Eurasian skylark, to assemble the sequence data and also included population RAD data from several species of larks. The study system is interesting from a conservation genetics perspective since there has been a monitoring of the population size for quite some time and is possible to directly contrast the census size with the diversity in the genome. They found much more genetic variation than expected from such a small population and also that substantial parts of the genome were sex-linked.

I think the manuscript in general is well-written and the figures are good. Despite having pretty limited within-population genetic data by today's standard (low coverage RAD sequence data), they manage to perform an impressive amount of population modelling.

I do think that the heterozygosity plots (Figure 3 and some of the supplemental figures) strongly support the existence of neosex chromosomes, which has also been suggested by earlier studies on lark karyotypes and by sequence data in other Sylvoidea species. The exact boundaries of these chromosome regions may be difficult to determine with this data set because coverage is limited and not all of the scaffolds could be aligned to the zebra finch genome. However, it seems convincing that recombination suppression is different between some of the focal species, particularly between the crested and raso lark on chromosome 3.

At times I feel that there is too much emphasis on the adaptive value of the neosex chromosomes in maintaining variation in the raso lark population (see for example lines 69-74, 420-421 and 450-451). To my understanding, this is not the general idea of how sex chromosomes form and are maintained, but it is rather through sexual antagonism (which is also mentioned briefly on lines

453-455). I think the authors should be a little bit more careful about the variation-buffering interpretation of these chromosomal changes.

One potential general concern of the study is that they use low coverage genetic data that is mapped to the reference of another species in the same genus. Using a reference genome of related species is indeed better than a de novo assembly of the RAD data and the authors also use several methods that could deal with low coverage sequence data, such as *angsd*. That said, I don't know if all of the methods they use are robust to the combination of very low coverage and cross-species mapping. One would expect that overall diversity in the raso lark could be somewhat underestimated due to the divergence from the reference genome, but I'm not sure if any specific skew would be predicted on the allele frequency spectrum and hence on the demographic inference. On top of the low coverage, RAD data also has some additional potential problems with allele dropouts and sequence duplicates originating from the PCR steps. Since this is single-end data and not paired-end data, there seems to be no easy way to identify and remove potential PCR duplicates. It would be good if the authors could speculate how much of a problem, if any, these things could be.

Related to this, I think it is important to more explicitly state the estimated divergence (average sequence divergence and/or divergence times in Myrs) between the studied species and specify the mapping rates of the reads from the different species. Table S2 provides the coverage of the mapped data, but no information is provided on how much trimmed sequence data there were for each sample in the beginning (i.e. sequence data that could potentially be mapped).

Figure 2 shows that when including females the neosex chromosomes have a large impact on the nucleotide diversity estimates in the raso lark. It is not clear to me though if any of the population history analyses, which were based on the inferred allele frequency spectrum, were also run on any of the subsets of the data (i.e. all samples excluding the sex-linked scaffolds or only males). Including the sex-linked SNPs and females must change the shape of the allele frequency spectrum quite a bit and could potentially have a large impact on the population modelling.

Minor comments

Figure S2: The heterozygosity plot of skylark populations from Eastern Russia and Mongolia looks much messier than the other skylark populations. Is it because there is more variation in coverage among these samples, lower coverage overall or something related to quality of these libraries? These samples should also be the ones that are most closely related to the reference genome. To clarify the origin of the samples please add a note in Table S2 if Russian samples are from the western or eastern part.

Line 121: For *allpaths*, please specify that the option “`–haploidfy=TRUE`” was used. This is at least what it says in the code provided on github. This is useful information for others who want to assemble their own data.

Line 145: Is it possible to clarify what a “mapping quality adjustment of 50” mean? The other settings (mapping and base quality) are more generic. Is it a default setting?

Lines 213-215: It seems to be a decent Illumina assembly, especially considering that the only mate pair library had an insert size of 3 kb. It would also be good to provide the number of scaffolds in the summary. To get a more independent quality measure, it would also be reasonable to use *busco* to quantify the number of conserved single copy genes, for example using the vertebrate or bird set.

Line 224: According to figure 3 there were three skylarks from the Netherlands that were removed because they had an excessive heterozygosity or homozygosity. Have they also been removed when calculating the nucleotide diversity for this population?

Line 424: The estimate of 42.2 million years must be associated with some uncertainty. Change to 42 Myrs or about 40 Myrs.

Lines 428-430: A linkage map is strictly not necessary to provide evidence of fusion events between these chromosomes. Some additional long-range scaffolding information (e.g. provided by optical maps or Hi-C) may be enough, at least for some of them. It is worth checking if there are any scaffolds with increased heterozygosity in females that map both to ancestral Z and chromosome 3/4A/5 in the zebra finch genome. It is unlikely, but I've seen the 4A-Z breakpoint in Illumina assemblies comparable to this in two other Sylvoidea species.

Lines 442-443: Alternatively, at least for chromosome 5, it is possible that not the entire chromosome was joined to the ancestral Z chromosome, as was the case for 4A.

Lines 444-445: Clarify that recombination suppression has expanded over a larger interval of chromosome 3.

Lines 483-485: 100 000 years seems to be an unrealistically low estimate for their colonization of the islands. This species is well diverged from its closest relatives. If the suggested scenario would be true, the presumably large source population from which they had spread from must have become extinct after this colonization and it is unclear why it would have. Unless the timing estimate is very misrepresentative (an order of magnitude too low), I think another biologically more plausible explanation must be used for the population bottleneck.

Line 485: Is this the total area of the islands or the area within them that would be suitable for raso lark. Or is it so that almost all area is suitable?

Decision letter (RSPB-2019-1102.R0)

18-Jun-2019

Dear Dr Dierickx:

I am writing to inform you that your manuscript RSPB-2019-1102 entitled "Neo-sex chromosomes and demography shape genetic diversity in the Critically Endangered Raso lark" has, in its current form, been rejected for publication in Proceedings B.

This action has been taken on the advice of referees, who have recommended that substantial revisions are necessary. With this in mind we would be happy to consider a resubmission, provided the comments of the referees are fully addressed. However please note that this is not a provisional acceptance.

Sincerely,
 Dr Daniel Costa
 mailto: proceedingsb@royalsociety.org

Associate Editor
 Board Member: 1
 Comments to Author:

This is essentially two stories on Raso larks, combined into one paper: the evolution of neo-sex chromosomes in larks and allies, and the demographic history of an island population of endangered bird species. Although the two topics are not immediately connected, the authors make good efforts to tie the two stories together.

The reviewers provide a number of recommendations for how the manuscript can be improved. From my own reading I had the same main issue as all reviewers, namely the farfetched idea of there being an adaptive value of evolving neo-sex chromosomes as means to maintain or increase genetic diversity. Certainly, the evidence for the deleterious effects associated with absence of recombination are overwhelming. There is therefore need for a major change in how the topic is presented throughout the manuscript, starting with the title and continuing in all sections (for example, on lines 69-82 in the Introduction).

The authors should also need to revise their analyses of how diversity/heterozygosity is estimated. To start with, they need to describe the assumptions they make on how the lark karyotypes are organized after the formation of neo-sex chromosomes. Critical in this respect is whether there are neo-Z chromosomes. I first thought so but then became uncertain when reading lines 428-431, where it is indicated that the authors don't know if there is a neo-Z (in each species). Specifically, what is the assumption behind estimating nucleotide diversity in males for chromosomes 3, 4a and 5 (are there two autosomal copies, two neo Z-linked copies, or some combination of these). And what is the assumption for estimation in females (one neo-Z and one neo-W, or something else)? This is related to that is unclear what is meant with the term (W chromosome) 'heterozygosity' in females. A non-recombining chromosome is haploid and there is no 'individual heterozygosity' (mentioned at several places).

Second, as pointed out by reviewers, the authors need to distinguish between diversity of sex-linked sequences (notably W-linked sequences), and divergence between paralogous chromosomal copies (i.e. gametologous sequences). The authors greatly overestimate diversity by including divergence between Z- and W-linked sequences (cf. previous point, assuming there is a neo-Z and a neo-W in females) in the diversity estimates. A re-analysis is needed here.

Minor comments

Lines 91-92. This is trivial and could be deleted.

Lines 171-172. Does this refer to >100 segregating sites or all sites, including monomorphic positions?

Lines 195. A generation time of 6.5 years for a small passerine birds seems very high. Cf. generation time estimates in other bird species.

Line 247. The observed heterozygosity is not that low compared to some other birds.

Figure 3 legend. Were the three mentioned skylark samples excluded from all analyses in the paper?

Hans Ellegren

Reviewer(s)' Comments to Author:

Referee: 1

Comments to the Author(s)

The manuscript "Neo-sex chromosomes and demography shape genetic diversity in the Critically Endangered Raso Lark" by Dierickx et al. aimed at understanding the determinants of genetic diversity in the small island population of Raso lark by specifically looking at the role of neo-sex chromosomes and demography.

The authors detect regions of the genome as neo-sex chromosomes using the higher observed heterozygosity in female individuals. They report genetic diversity for the Raso lark and argue that the observed diversity is higher than what would be expected for an island species of such a small population size. They go on to show that by excluding the neo-sex chromosomes, genetic diversity is halved in the Raso lark and consider neo-sex chromosomes as a source of genetic variation. To further explain the observed level of genetic diversity, they conduct demographic modeling in which they show that demographic history of the species is best explained by incorporating both population contraction and an ancient expansion leading to an increase in rare variants.

The authors produced a draft genome assembly for the Eurasian skylark and RAD-seq reads for 78 individuals of 4 lark species which can be used as a valuable resource for further studies on larks and other avian comparative studies.

However, my main concern regarding this manuscript is that the elevated heterozygosity reported in Raso larks is obtained by calculating heterozygosity in neo-sex chromosomes in females. I think this is not correct since the neo-Z and neo-W chromosomes are diverged from one another and this means that the divergence between neo-Z and neo-W chromosomes contribute to the heterozygosity, i.e., $d_{xy} = 4N_{eu} + 2ut$ which leads to an overestimation of the effective population size. Moreover, authors argue that recombination suppression has retained genetic variation while this is a known fact that recombination suppression leads to the reduction in genetic diversity. Since the role of neo-sex chromosomes in maintaining genetic variation is a core part of the paper, I am afraid that this result has been incorrectly obtained by using the divergence between neo-Z and neo-W in females.

I think this manuscript is an interesting work for the Proceedings of the Royal Society B, however, it requires major revisions before it can be considered for publication as detailed in the comments below:

Introduction:

1. Line 55: Reference 12 gives a mixed message about the genetic diversity and N_e of island species. According to reference 12, some island species show reduced genetic diversity and some show no differences compared to mainland species. I think it would be more comprehensive if both aspects are mentioned in the introduction.
2. Line 70: A reference is required at the end of the sentence: "Previous work suggests that a change to genome architecture might buffer Raso larks from genetic diversity loss."
3. Lines 71: I think a schematic figure of the *Alauda* neo-sex chromosomes in the introduction would help the reader to follow the rest of the paper more easily. It would be helpful if the figure shows the recombining and non-recombining part of the neo-sex chromosomes together with the segment used for measuring genetic diversity in females.
4. Line 71 to 74: It is stated that "Cessation of recombination on neo-sex chromosomes could represent a source of heterozygosity, because females (the heterogametic sex) could retain distinct alleles at homologous loci on their neo-Z and neo-W chromosomes." I am confused about this statement. By homologous loci on neo-Z and neo-W chromosomes, are the authors referring to the so-called gametologous genes (homologous genes on the non-recombining sex chromosomes that are not yet degenerated from the W)? First, cessation of recombination leads to a decreased effective population size and therefore, reduction in heterozygosity. Second, due to the lack of recombination, the neo-Z and neo-W sequences have diverged from one another. All females are heterozygous not because two alleles are segregating in the population but because all W sequences are fixed for one allele and all Z sequences for the other allele. This is the result of divergence of two sequences since the cessation of recombination prior to the split of the species studied. It is not therefore correct to measure heterozygosity for females using neo-Z and neo-W sequences since the Tajim's π should tell us about the amount of genetic diversity since the most recent common ancestor of the species of interest. Please see for example the analyses in Bachtrog and Charlesworth 2000 regarding neo-sex chromosomes.

Methods:

5. Line 95: It is very good that all scripts are provided in the github page. A master script is however necessary which explains the order scripts should be run. For example, if a researcher downloads the raw data of this study from ENA and wants to reproduce the study, that master script should provide the steps in the pipeline.
6. I suggest a reordering of the paragraphs in the Methods section as follows:
 - Sample collection
 - Whole genome sequencing and assembly
 - RAD library preparation
 - Sequence processing and alignment
 - Pseudo-chromosomal assembly
 - Relatedness
 - "Allele frequency spectra and genetic diversity" merged with "Proportion of heterozygous sites across the genome" into one subheading
 - Demographic inference

Results:

7. Since the sequence of neo-sex chromosomes of several species is available here, it is possible to get an estimate of the timing of recombination suppression by calculating d_s . The authors could add this analysis to their manuscript to improve the inference of the timing of recombination suppression events.

8. Line 217: What is the percentage of mapped reads from Raso lark to the Eurasian Skylark assembly?

9. Line 223: Please report Tajima's π and Watterson θ for each case in males and separately for autosomes and neo-Z chromosomes.

10. Line 225: Genetic diversity in the Eurasian skylark from the Netherlands is 0.0097. This tells us about 1 heterozygous SNP in almost every 100 base pairs. This is really a high level of genetic diversity. What is the range of genetic diversity across other studied passerines? How does this high level of genetic diversity might have affected assembly construction? Did the high level of heterozygosity provide some challenges for the mapping of reads from Raso larks to the Eurasian Skylark assembly?

11. Line 226: Equation $\theta = 4N_e\mu$ is used to obtain N_e from genetic diversity. The genetic diversity used is 0.0018 which contains the heterozygosity obtained from neo-sex chromosomes in females. I think this might not be correct since the N_e obtained from $\theta = 4N_e\mu$ is the time to the most recent common ancestor of the sample. The diversity obtained from the neo-sex chromosomes in females is actually $d_{xy} = 4N_e\mu + 2ut$. This means that heterozygosity is overestimated by taking into account divergence between sequences leading to an overestimation of N_e . I think higher female heterozygosity must only be used as a method to detect suppressed recombination. Then to calculate heterozygosity in Raso lark, neo-sex chromosomes must be masked and only diversity of autosomes should be used. If sex chromosomes are to be used, then diversity should be calculated for the Z chromosome and W chromosome separately and in calculating N_e , their relative numbers for every male and female should be taken into account. For example, in a standard case, there are 3 Z and 1 W for every 4 autosomes for a male and a female in the population.

12. Line 245: The heterozygosity measured between neo-Z and neo-W chromosomes in females cannot be compared with heterozygosity between two neo-Z in males. The reason is that the heterozygosity obtained in females is actually divergence between the Z and W sequences. Once recombination is suppressed, the Z and W chromosomes start to differentiate from one another. This is similar to two subpopulations between which migration stops. To calculate genetic diversity, one should use only the Z sequences or the W sequences. Therefore, while elevated heterozygosity in females can be used as a technical way to find regions with suppressed recombination, it does not mean that we have maintenance of diversity due to recombination suppression.

13. Lines 249 to 258: From "Previously, only party of ..." are not the results of this paper or are the discussion regarding the result, I suggest to move this part to the discussion.

14. Lines 275 to 279: From "A similar analysis ..." to "... that excludes chromosomes 5 [38]" are not results of this paper and I suggest to move this section to the discussion.

15. Line 303: This π without the neo-sex chromosomes is the correct π and this should be reported as the heterozygosity in Raso Larks.

16. Line 304: "Recombination suppression across the neo-sex chromosomes therefore does indeed contribute to the maintenance of genetic variation in the Raso lark.". This sentence is not correct. Recombination suppression has led to a decrease in genetic variation, as mentioned before, one cannot use divergence between two sequences to calculate heterozygosity. I suggest to remove this sentence.

17. Line 312: It is stated that diversity difference between the Raso lark and the Eurasian skylark "is far greater than expected given the difference in current population sizes between the two species". This is not surprising because it is the N_e that matters not the census population size.

For example, census population size of humans is about 8 billion, that of *Pan troglodytes* is in the order of tens of thousands only, yet genetic diversity in chimps is higher than that of humans due to their higher N_e . What is “expected” as a relationship between the census size and genetic diversity?

18. Line 391: I suggest that the Relatedness section to be moved to the beginning of results as it is more of a quality check and it is important to be presented before the estimates of diversity.

Discussion:

19. Line 413: “... this difference is smaller than expected.”. What is the expected difference?

20. Line 436: “... 1-1.5% divergence”. This is a result not reported in the result section. Please add to both Methods and Results the text regarding the calculation of divergence between the neo-Z and neo-W chromosomes.

21. Line 451: “... has been favored by selection following the loss of genetic diversity.” Please clarify this sentence.

Line 471: “... level of divergence between the neo-Z and neo-W is similar in all three *Aluade* species.”. This is a result not presented in the result section. If known from previous work, please cite the reference.

Referee: 2

Comments to the Author(s)

Dear Editor:

Dierickx et al. profiled the genetic diversity of different lark population, particularly the endangered Raso Lark concentrated at two islands of Africa, regarding their chromosome-wide polymorphism level. They found signatures of neo-sex chromosomes, which are different between different skylark and lark populations. Then they tried different methods to fit the historical change of population size to the unexpectedly high polymorphism level as compared to its reported population size nowadays. Maybe because the population contraction is too recent, the current data cannot find a clear signature for the recent population contraction. I have several comments about the paper:

1. It is a bit confusing to state that neo-sex chromosomes maintain the heterozygosities in Raso lark, as the author mentioned in several places in the paper. As first neo-sex chromosomes probably formed by genetic drift in this case, due to the population size reduction. And generally, the differences between neo-Z and neo-W are expected to be excessive deleterious mutations that are fixed on the neo-W, rather than something that the population would like to maintain.
2. Given the current chromosomal sequences are using zebra finch as the reference (as the author also pointed out), it is very likely that the current fusion order in Figure 4 would be subjected to change. First, it is difficult to imagine that chr5, which is more distant from chrW than chr3, to first suppress recombination. Because one would expect sexual antagonistic selection would first accumulate nearby the W first, so that they are more likely to be inherited in only one sex. Second, there is a dip of heterozygosity in most populations between the first part and second part of chr3 (Figure 3, Raso lark). Is this caused by sequencing/assembly gaps? And also, is the female heterozygosity level of the first part of chr3 significantly different from those of the second part? This is important if they actually formed at separate time points. Another question, does the author have some explanation for the elevated heterozygosity level of male on the Z chromosome in Eurasian skylark? This is too obvious to be ignored.

3. Regarding the demographic analyses, I would think the most straightforward way is collect data from nearby islands that used to be inhabited by Raso lark? Would this be possible from the museum specimen?

Referee: 3

Comments to the Author(s)

Dierickx et al. used reduced representation (RAD) sequencing to explore genetic diversity and the population history of raso larks. They used the reference from a close relative, the Eurasian skylark, to assemble the sequence data and also included population RAD data from several species of larks. The study system is interesting from a conservation genetics perspective since there has been a monitoring of the population size for quite some time and is possible to directly contrast the census size with the diversity in the genome. They found much more genetic variation than expected from such a small population and also that substantial parts of the genome were sex-linked.

I think the manuscript in general is well-written and the figures are good. Despite having pretty limited within-population genetic data by today's standard (low coverage RAD sequence data), they manage to perform an impressive amount of population modelling.

I do think that the heterozygosity plots (Figure 3 and some of the supplemental figures) strongly support the existence of neosex chromosomes, which has also been suggested by earlier studies on lark karyotypes and by sequence data in other Sylvoidea species. The exact boundaries of these chromosome regions may be difficult to determine with this data set because coverage is limited and not all of the scaffolds could be aligned to the zebra finch genome. However, it seems convincing that recombination suppression is different between some of the focal species, particularly between the crested and raso lark on chromosome 3.

At times I feel that there is too much emphasis on the adaptive value of the neosex chromosomes in maintaining variation in the raso lark population (see for example lines 69-74, 420-421 and 450-451). To my understanding, this is not the general idea of how sex chromosomes form and are maintained, but it is rather through sexual antagonism (which is also mentioned briefly on lines 453-455). I think the authors should be a little bit more careful about the variation-buffering interpretation of these chromosomal changes.

One potential general concern of the study is that they use low coverage genetic data that is mapped to the reference of another species in the same genus. Using a reference genome of related species is indeed better than a de novo assembly of the RAD data and the authors also use several methods that could deal with low coverage sequence data, such as angsd. That said, I don't know if all of the methods they use are robust to the combination of very low coverage and cross-species mapping. One would expect that overall diversity in the raso lark could be somewhat underestimated due to the divergence from the reference genome, but I'm not sure if any specific skew would be predicted on the allele frequency spectrum and hence on the demographic inference. On top of the low coverage, RAD data also has some additional potential problems with allele dropouts and sequence duplicates originating from the PCR steps. Since this is single-end data and not paired-end data, there seems to be no easy way to identify and remove potential PCR duplicates. It would be good if the authors could speculate how much of a problem, if any, these things could be.

Related to this, I think it is important to more explicitly state the estimated divergence (average sequence divergence and/or divergence times in Myrs) between the studied species and specify the mapping rates of the reads from the different species. Table S2 provides the coverage of the mapped data, but no information is provided on how much trimmed sequence data there were for each sample in the beginning (i.e. sequence data that could potentially be mapped).

Figure 2 shows that when including females the neosex chromosomes have a large impact on the nucleotide diversity estimates in the raso lark. It is not clear to me though if any of the population history analyses, which were based on the inferred allele frequency spectrum, were also run on any of the subsets of the data (i.e. all samples excluding the sex-linked scaffolds or only males). Including the sex-linked SNPs and females must change the shape of the allele frequency spectrum quite a bit and could potentially have a large impact on the population modelling.

Minor comments

Figure S2: The heterozygosity plot of skylark populations from Eastern Russia and Mongolia looks much messier than the other skylark populations. Is it because there is more variation in coverage among these samples, lower coverage overall or something related to quality of these libraries? These samples should also be the ones that are most closely related to the reference genome. To clarify the origin of the samples please add a note in Table S2 if Russian samples are from the western or eastern part.

Line 121: For allpaths, please specify that the option “`–haploidfy=TRUE`” was used. This is at least what it says in the code provided on github. This is useful information for others who want to assemble their own data.

Line 145: Is it possible to clarify what a “mapping quality adjustment of 50” mean? The other settings (mapping and base quality) are more generic. Is it a default setting?

Lines 213-215: It seems to be a decent Illumina assembly, especially considering that the only mate pair library had an insert size of 3 kb. It would also be good to provide the number of scaffolds in the summary. To get a more independent quality measure, it would also be reasonable to use busco to quantify the number of conserved single copy genes, for example using the vertebrate or bird set.

Line 224: According to figure 3 there were three skylarks from the Netherlands that were removed because they had an excessive heterozygosity or homozygosity. Have they also been removed when calculating the nucleotide diversity for this population?

Line 424: The estimate of 42.2 million years must be associated with some uncertainty. Change to 42 Myrs or about 40 Myrs.

Lines 428-430: A linkage map is strictly not necessary to provide evidence of fusion events between these chromosomes. Some additional long-range scaffolding information (e.g. provided by optical maps or Hi-C) may be enough, at least for some of them. It is worth checking if there are any scaffolds with increased heterozygosity in females that map both to ancestral Z and chromosome 3/4A/5 in the zebra finch genome. It is unlikely, but I've seen the 4A-Z breakpoint in Illumina assemblies comparable to this in two other Sylvoidea species.

Lines 442-443: Alternatively, at least for chromosome 5, it is possible that not the entire chromosome was joined to the ancestral Z chromosome, as was the case for 4A.

Lines 444-445: Clarify that recombination suppression has expanded over a larger interval of chromosome 3.

Lines 483-485: 100 000 years seems to be an unrealistically low estimate for their colonization of the islands. This species is well diverged from its closest relatives. If the suggested scenario would be true, the presumably large source population from which they had spread from must have become extinct after this colonization and it is unclear why it would have. Unless the timing estimate is very misrepresentative (an order of magnitude too low), I think another biologically more plausible explanation must be used for the population bottleneck.

Line 485: Is this the total area of the islands or the area within them that would be suitable for raso lark. Or is it so that almost all area is suitable?

Author's Response to Decision Letter for (RSPB-2019-1102.R0)

See Appendix A.

RSPB-2019-2613.R0

Review form: Reviewer 1

Recommendation

Accept with minor revision (please list in comments)

Scientific importance: Is the manuscript an original and important contribution to its field?

Good

General interest: Is the paper of sufficient general interest?

Good

Quality of the paper: Is the overall quality of the paper suitable?

Good

Is the length of the paper justified?

Yes

Should the paper be seen by a specialist statistical reviewer?

No

Do you have any concerns about statistical analyses in this paper? If so, please specify them explicitly in your report.

No

It is a condition of publication that authors make their supporting data, code and materials available - either as supplementary material or hosted in an external repository. Please rate, if applicable, the supporting data on the following criteria.

Is it accessible?

Yes

Is it clear?

Yes

Is it adequate?

Yes

Do you have any ethical concerns with this paper?

No

Comments to the Author

The authors have put a considerable effort and made major revisions to the manuscript which has greatly improved the work. The main point of including ZW divergence in the overall diversity is now corrected, the methods read very clearly now and the analyses steps and codes are described and provided in the github repository.

Please find below some minor but I think important comments to be addressed:

(Page and line numbers are provided based on the TrackChanges version of the manuscript)

Results:

Page 8 - Line 273: "The gametologues have evidently not diverged to the point that sequence homology has been eliminated."

The term "homology" has been used incorrectly in this sentences. Homology reflects the evolutionary history. Two characters or genes are homologous if they share a common ancestor and if they are homologous, that homology can never be eliminated. In the case of Z and W genes, since they share a common ancestor, they are always homologous regardless of the degree of degeneration. Instead of homology, "sequence identity" or "sequence similarity" can be used.

Page 14 - Line 467: "... indicate fairly strong homology ..."

Same comment as above. Homology is a 0 or 1 situation, it's either there or not. You could modify it simply to "... fairly strong sequence identity/similarity ..."

Discussion:

Page 15 - Line 516: "... rule out an increase in mutation rate because the level of divergence between the neo-Z and neo-W is similar ..."

This sentence seems contradictory to me to what the authors argue throughout the paper about female heterozygosity. It is mainly argued throughout the paper that the higher female heterozygosity is a result of maintenance of ancestral genetic variability. However, the message in line 516 is that the observed "female heterozygosity" is actually due to divergence of the W sequence from the Z as a result of their lineage-specific mutations after recombination suppression and not the ancestral polymorphism. It would be great if the authors could clarify this a bit further.

Supplementary material page 10: I could not find a separate Excel spreadsheet on Sample information (S2) in the attached files.

Review form: Reviewer 2**Recommendation**

Reject – article is not of sufficient interest (we will consider a transfer to another journal)

Scientific importance: Is the manuscript an original and important contribution to its field?

Acceptable

General interest: Is the paper of sufficient general interest?

Acceptable

Quality of the paper: Is the overall quality of the paper suitable?

Good

Is the length of the paper justified?

Yes

Should the paper be seen by a specialist statistical reviewer?

No

Do you have any concerns about statistical analyses in this paper? If so, please specify them explicitly in your report.

No

It is a condition of publication that authors make their supporting data, code and materials available - either as supplementary material or hosted in an external repository. Please rate, if applicable, the supporting data on the following criteria.

Is it accessible?

Yes

Is it clear?

Yes

Is it adequate?

Yes

Do you have any ethical concerns with this paper?

No

Comments to the Author

Dear Editor:

When reviewing this paper, I found another group has published the very similar story on the same journal on the larks' sex chromosomes (<https://royalsocietypublishing.org/doi/abs/10.1098/rspb.2019.2051>). Although this work has a slightly different focus on the population dynamics, but the major point, that several chromosome segments have probably fused to the ancestral sex chromosomes of birds in larks, is the same as the published paper. While the published paper produced more data with whole genome sequencing, and provided more detailed analyses on the age of fusion, i.e., the evolutionary strata of the lark sex chromosomes, and then compared these fused chromosomes in terms of other vertebrate sex chromosomes. This draws a broader attention than the current paper that I am reviewing. Thus I am afraid I have to recommend rejection of the paper.

Review form: Reviewer 3

Recommendation

Accept with minor revision (please list in comments)

Scientific importance: Is the manuscript an original and important contribution to its field?

Acceptable

General interest: Is the paper of sufficient general interest?

Good

Quality of the paper: Is the overall quality of the paper suitable?

Good

Is the length of the paper justified?

Yes

Should the paper be seen by a specialist statistical reviewer?

No

Do you have any concerns about statistical analyses in this paper? If so, please specify them explicitly in your report.

No

It is a condition of publication that authors make their supporting data, code and materials available - either as supplementary material or hosted in an external repository. Please rate, if applicable, the supporting data on the following criteria.

Is it accessible?

Yes

Is it clear?

Yes

Is it adequate?

Yes

Do you have any ethical concerns with this paper?

No

Comments to the Author

The authors have addressed most of the issues that I and other reviewers raised for the previous version of the manuscript. I think that the methods and the results are clearer now and that the genetic analyses are more appropriate.

They have also modified the text to tone down the hypothesis that the neosex chromosomes may have a selective benefit of maintaining variation in females. It is now emphasized that this mechanism is unlikely to have been an initial selection pressure for the formation of the neosex chromosomes. However, they keep the idea that the presence of neosex chromosomes may inadvertently have contributed to fitness in females through the maintenance of pre-existing allelic variation that predates the suppression of recombination between the gametologues.

I don't think this is a probable scenario. Based on the recent study of Sigeman et al. (2019) (Repeated sex chromosome evolution in vertebrates supported by expanded avian sex chromosomes. Proc. R. Soc. B 286), which some of the authors here are included in, the recombination cessation appears to have occurred a very long time ago, with the most recent event occurring at least at the time before the skylark and the raso lark diverged (estimated 6 Myrs or more) and several probably occurring more than 15 Myrs ago. With these time scales I would expect that most of the divergence between the Z and W copies has happened after the recombination cessation and that a number of these changes will have a deleterious effect. The 1 % divergence reported here seems like it could be an underestimate of how different the gametologues really are. On top of that one would also expect to see some variation in divergence between the evolutionary strata associated with the different chromosome fusions, which is also

reported in the Sigeman et al. study. Furthermore, there is no direct measurement of how degenerate the W-linked sequences are in this study. Just because there still is homology between the W and Z copies, it doesn't rule out that there hasn't already been deleterious changes.

To me there seems to be no real evidence supporting the variation-buffering hypothesis, but it rather seems to originate from the observation that this is a small endangered population and that pooling reads from the divergent Z and W copies creates a lot of additional "heterozygosity", which presumably would be beneficial for the population. None of these neosex chromosomes are even unique to this species and are shared by the abundant skylark.

That said, I want to emphasize that this is their interpretation of their data and that this doesn't disqualify the results. However, I still wonder if it would not be more appropriate to tone down this hypothesis even more or even removing it, because it doesn't seem reasonable.

Decision letter (RSPB-2019-2613.R0)

17-Dec-2019

Dear Dr Dierickx:

Your manuscript has now been peer reviewed and the reviews have been assessed by an Associate Editor. The reviewers' comments (not including confidential comments to the Editor) and the comments from the Associate Editor are included at the end of this email for your reference. As you will see, the reviewer are mixed and have raised some concerns with your manuscript. I am willing to give you a final opportunity to respond to these comments before we make a final decision.

Research ethics:

Use of animals and field studies:

Please submit a copy of your revised paper within three weeks. If we do not hear from you within this time your manuscript will be rejected. If you are unable to meet this deadline please let us know as soon as possible, as we may be able to grant a short extension.

Best wishes,
Dr Daniel Costa
<mailto:proceedingsb@royalsociety.org>

Reviewer(s)' Comments to Author:

Referee: 1

Comments to the Author(s).

The authors have put a considerable effort and made major revisions to the manuscript which has greatly improved the work. The main point of including ZW divergence in the overall diversity is now corrected, the methods read very clearly now and the analyses steps and codes are described and provided in the github repository.

Please find below some minor but I think important comments to be addressed:

(Page and line numbers are provided based on the TrackChanges version of the manuscript)

Results:

Page 8 - Line 273: "The gametologues have evidently not diverged to the point that sequence homology has been eliminated."

The term "homology" has been used incorrectly in this sentences. Homology reflects the evolutionary history. Two characters or genes are homologous if they share a common ancestor and if they are homologous, that homology can never be eliminated. In the case of Z and W genes, since they share a common ancestor, they are always homologous regardless of the degree of degeneration. Instead of homology, "sequence identity" or "sequence similarity" can be used.

Page 14 - Line 467: "... indicate fairly strong homology ..."

Same comment as above. Homology is a 0 or 1 situation, it's either there or not. You could modify it simply to "... fairly strong sequence identity/similarity ..."

Discussion:

Page 15 - Line 516: "... rule out an increase in mutation rate because the level of divergence between the neo-Z and neo-W is similar ..."

This sentence seems contradictory to me to what the authors argue throughout the paper about female heterozygosity. It is mainly argued throughout the paper that the higher female heterozygosity is a result of maintenance of ancestral genetic variability. However, the message in line 516 is that the observed "female heterozygosity" is actually due to divergence of the W sequence from the Z as a result of their lineage-specific mutations after recombination suppression and not the ancestral polymorphism. It would be great if the authors could clarify this a bit further.

Supplementary material page 10: I could not find a separate Excel spreadsheet on Sample information (S2) in the attached files.

Referee: 2

Comments to the Author(s).

Dear Editor:

When reviewing this paper, I found another group has published the very similar story on the same journal on the larks' sex chromosomes (<https://royalsocietypublishing.org/doi/abs/10.1098/rspb.2019.2051>). Although this work has a

slightly different focus on the population dynamics, but the major point, that several chromosome segments have probably fused to the ancestral sex chromosomes of birds in larks, is the same as the published paper. While the published paper produced more data with whole genome sequencing, and provided more detailed analyses on the age of fusion, i.e., the evolutionary strata of the lark sex chromosomes, and then compared these fused chromosomes in terms of other vertebrate sex chromosomes. This draws a broader attention than the current paper that I am reviewing. Thus I am afraid I have to recommend rejection of the paper.

Referee: 3

Comments to the Author(s).

The authors have addressed most of the issues that I and other reviewers raised for the previous version of the manuscript. I think that the methods and the results are clearer now and that the genetic analyses are more appropriate.

They have also modified the text to tone down the hypothesis that the neosex chromosomes may have a selective benefit of maintaining variation in females. It is now emphasized that this mechanism is unlikely to have been an initial selection pressure for the formation of the neosex chromosomes. However, they keep the idea that the presence of neosex chromosomes may inadvertently have contributed to fitness in females through the maintenance of pre-existing allelic variation that predates the suppression of recombination between the gametologues.

I don't think this is a probable scenario. Based on the recent study of Sigeman et al. (2019) (Repeated sex chromosome evolution in vertebrates supported by expanded avian sex chromosomes. *Proc. R. Soc. B* 286), which some of the authors here are included in, the recombination cessation appears to have occurred a very long time ago, with the most recent event occurring at least at the time before the skylark and the raso lark diverged (estimated 6 Myrs or more) and several probably occurring more than 15 Myrs ago. With these time scales I would expect that most of the divergence between the Z and W copies has happened after the recombination cessation and that a number of these changes will have a deleterious effect. The 1 % divergence reported here seems like it could be an underestimate of how different the gametologues really are. On top of that one would also expect to see some variation in divergence between the evolutionary strata associated with the different chromosome fusions, which is also reported in the Sigeman et al. study. Furthermore, there is no direct measurement of how degenerate the W-linked sequences are in this study. Just because there still is homology between the W and Z copies, it doesn't rule out that there hasn't already been deleterious changes.

To me there seems to be no real evidence supporting the variation-buffering hypothesis, but it rather seems to originate from the observation that this is a small endangered population and that pooling reads from the divergent Z and W copies creates a lot of additional "heterozygosity", which presumably would be beneficial for the population. None of these neosex chromosomes are even unique to this species and are shared by the abundant skylark.

That said, I want to emphasize that this is their interpretation of their data and that this doesn't disqualify the results. However, I still wonder if it would not be more appropriate to tone down this hypothesis even more or even removing it, because it doesn't seem reasonable.

Author's Response to Decision Letter for (RSPB-2019-2613.R0)

See Appendix B.

RSPB-2019-2613.R1 (Revision)

Review form: Reviewer 1

Recommendation

Accept as is

Scientific importance: Is the manuscript an original and important contribution to its field?

Good

General interest: Is the paper of sufficient general interest?

Good

Quality of the paper: Is the overall quality of the paper suitable?

Good

Is the length of the paper justified?

Yes

Should the paper be seen by a specialist statistical reviewer?

No

Do you have any concerns about statistical analyses in this paper? If so, please specify them explicitly in your report.

No

It is a condition of publication that authors make their supporting data, code and materials available - either as supplementary material or hosted in an external repository. Please rate, if applicable, the supporting data on the following criteria.

Is it accessible?

Yes

Is it clear?

Yes

Is it adequate?

Yes

Do you have any ethical concerns with this paper?

No

Comments to the Author

My comments are addressed by the authors and I do not have further comments on the manuscript.

Review form: Reviewer 2

Recommendation

Accept as is

Scientific importance: Is the manuscript an original and important contribution to its field?
Acceptable

General interest: Is the paper of sufficient general interest?
Good

Quality of the paper: Is the overall quality of the paper suitable?
Acceptable

Is the length of the paper justified?
Yes

Should the paper be seen by a specialist statistical reviewer?
No

Do you have any concerns about statistical analyses in this paper? If so, please specify them explicitly in your report.
No

It is a condition of publication that authors make their supporting data, code and materials available - either as supplementary material or hosted in an external repository. Please rate, if applicable, the supporting data on the following criteria.

Is it accessible?
Yes

Is it clear?
Yes

Is it adequate?
Yes

Do you have any ethical concerns with this paper?
Yes

Comments to the Author

I previously raised the concern that the authors pushed too hard on the variation-buffering hypothesis of the neosex chromosomes. In the revised manuscript, they have removed these sentences from the abstract, introduction and results. They now only mention the hypothesis in the discussion and emphasize that it was put forward by an earlier study. I think that these changes are good enough.

Decision letter (RSPB-2019-2613.R1)

10-Feb-2020

Dear Dr Dierickx

I am pleased to inform you that your manuscript entitled "Genetic diversity, demographic history and neo-sex chromosomes in the Critically Endangered Raso lark" has been accepted for publication in Proceedings B.

You can expect to receive a proof of your article from our Production office in due course, please

check your spam filter if you do not receive it. PLEASE NOTE: you will be given the exact page length of your paper which may be different from the estimation from Editorial and you may be asked to reduce your paper if it goes over the 10 page limit.

Open Access

Paper charges

Sincerely,

Dr Daniel Costa

Appendix A

Responses to Reviewers

We are grateful to the editor and reviewers for their insightful and constructive comments. We have extensively revised our manuscript to address all concerns. The major changes are as follows:

- 1. We have rearranged the order of the results so as to deal with the neo-sex chromosomes first and remove any doubt that we have incorrectly estimated levels of diversity.**
- 2. We have revised our terminology to clarify that proportions of heterozygous genotypes on the neo-sex chromosomes represent divergence between gametologues and not heterozygosity in the conventional sense.**
- 3. We have clarified that the hypothesis that the neo-sex chromosomes may protect functional genetic variation from loss due to drift came from an earlier study. However, we still conclude that an inadvertent benefit may be plausible, because the gametologues retain nearly 99% identity and likely retain standing genetic variation that pre-dates the recent onset of suppressed recombination, and may therefore be functional.**
- 4. In revising the analyses we re-called all genotypes with GATK HaplotypeCaller, which have lower error rates. This meant that all analyses have been redone. The qualitative results are unchanged, but the numbers have been slightly changed.**
- 5. The title has changed to “*Neo-sex chromosomes, genetic diversity and demographic history in the Critically Endangered Raso lark*”.**

We feel that this revision process has greatly improved our paper.

Responses to all comments are given in BOLD below.

Associate Editor
Board Member: 1

Comments to Author:

This is essentially two stories on Raso larks, combined into one paper: the evolution of neo-sex chromosomes in larks and allies, and the demographic history of an island population of endangered bird species. Although the two topics are not immediately connected, the authors make good efforts to tie the two stories together.

The reviewers provide a number of recommendations for how the manuscript can be improved. From my own reading I had the same main issue as all reviewers, namely the farfetched idea of there being an adaptive value of evolving neo-sex chromosomes as means to maintain or increase genetic diversity. Certainly, the evidence for the deleterious effects associated with absence of recombination

are overwhelming. There is therefore need for a major change in how the topic is presented throughout the manuscript, starting with the title and continuing in all sections (for example, on lines 69-82 in the Introduction).

We have re-written large parts of the manuscript to address this concern. We note that the potential adaptive value of the suppressed recombination in maintaining diversity was proposed previously, in a paper entitled “Widespread translocation from autosomes to sex chromosomes preserves genetic variability in an endangered lark” (Brooke et al. 2010). We therefore cannot ignore the possibility, but now present it as simply a hypothesis, and make it clear that it comes from previous work.

However, we do believe that the hypothesis has some support. It is likely that some of the divergence between the neo-sex chromosome gametologues reflects pre-existing allelic variation that predates the suppression of recombination and has effectively been trapped and prevented from loss due to drift. This is analogous to the retention of heterozygosity in functionally asexual hybrid plants. We agree that the neo-W will eventually degenerate, but this is evidently not yet the case: it shares nearly 99% identity with the neo-Z in the regions of suppressed recombination, and therefore likely still contains functional ancestral alleles that will be absent in males. We make this point in the revised version as follows:

“Nonetheless suppressed recombination might inadvertently result in the maintenance of pre-existing allelic variation between the gametologues that could theoretically contribute to functional diversity and therefore fitness in females [17]. This is analogous to the retention of heterozygosity in functionally asexual hybrid plants [47], except that here it applies only to part of the genome, and only to females.”

We certainly do not intend to propose that any potential benefit of retained allelic variation in females has been a selective driver that has driven the formation of the neo-sex chromosomes, but we do believe that it may be an inadvertent consequence, and this possibility should be considered in future work.

The authors should also need to revise their analyses of how diversity/heterozygosity is estimated. To start with, they need to describe the assumptions they make on how the lark karyotypes are organized after the formation of neo-sex chromosomes. Critical in this respect is whether there are neo-Z chromosomes. I first thought so but then became uncertain when reading lines 428-431, where it is indicated that the authors don't know if there is a neo-Z (in each species).

Specifically, what is the assumption behind estimating nucleotide diversity in males for chromosomes 3, 4a and 5 (are there two autosomal copies, two neo Z-linked copies, or some combination of these). And what is the assumption for estimation in females (one neo-Z and one neo-W, or something else)? This is

related to that is unclear what is meant with the term (W chromosome) 'heterozygosity' in females.

We thank the reviewers and editor for showing that we hadn't clearly articulated our model for the neo-Z and neo-W formations. We have addressed this concern by clarifying the text on lines

There is a neo-Z in the sense that males carry two homologous copies of chromosomes 3, 4a and 5, and females carry two gametologous copies of these same chromosomes. The fact that reads map and we can call heterozygous genotypes in females indicates that the gametologues are not highly diverged, and still retain strong identity of nearly 99%.

It is unclear whether the copy of each of these autosomes that is not behaving as a W is physically fused to the ancestral W, and likewise whether the Z copy is fused to the ancestral Z, and we now make that uncertainty clear. Cases of multiple unfused sex chromosomes are known in other taxa (Sahara et al. 2012, Chromosome Research, Rens et al. 2007 Genome Biology), but in our case the visual enlargement of the lark sex chromosomes does suggest fusions.

A non-recombining chromosome is haploid and there is no 'individual heterozygosity' (mentioned at several places).

This statement is correct for the case of ancient sex chromosomes that have diverged strongly, but is not entirely true for those that have undergone recombination suppression very recently, nor for pseudo-autosomal regions where recombination persists. We accept that this is distinct from heterozygosity in the conventional sense, and we now refer to the "density of heterozygous genotypes". We have clarified the logic in the paper as follows:

"As females carry a single copy of the Z, they should have no heterozygous sites on this chromosome (with the exception of repetitive elements at which mis-mapping can occur, and possibly pseudo-autosomal regions where the W and Z retain homology). The same should be true for neo-sex chromosomes, unless recombination suppression is recent or incomplete, such that the gametologues retain a high level of homology. This would allow RAD-seq reads from the neo-W to map to scaffolds representing the neo-Z (there should be no neo-W scaffolds in our male reference genome). This could potentially lead to an elevation (rather than a reduction) in the density of heterozygous genotype calls in females. Indeed, we observe strong elevations in the density of heterozygous genotypes across large portions of chromosomes 3, 4a and 5 in all female Raso larks and not in males (Figure 2A). This is consistent with the formation of neo-sex chromosomes and subsequent recombination suppression involving these three autosomes. The gametologues have evidently not diverged

to the point that sequence homology has been eliminated. Indeed they retain around 99% similarity (1.15 differences per 100 bp on average)

Second, as pointed out by reviewers, the authors need to distinguish between diversity of sex-linked sequences (notably W-linked sequences), and divergence between paralogous chromosomal copies (i.e. gametologous sequences).

In the revised manuscript, we now refer to neo-Z and neo-W as “gametologous”. Thank you for the suggestion.

The authors greatly overestimate diversity by including divergence between Z- and W-linked sequences (cf. previous point, assuming there is a neo-Z and a neo-W in females) in the diversity estimates. A re-analysis is needed here.

The editor is absolutely correct that including the divergence between gametologues in genetic diversity calculations to understand effective population size would be flawed. That is a key point in our paper. Our original manuscript confused this message by only revealing the existence of the neo-sex chromosomes after first presenting a diversity estimate. That was how we discovered them, but we accept that the correct order is to first describe the neo-sex chromosomes and then compute diversity correctly by excluding the regions of suppressed recombination.

Minor comments

Lines 91-92. This is trivial and could be deleted.

The final sentence in the abstract has been reworded as:

“Our findings show how genome-wide approaches to study endangered species can help avoid confounding effects of genome architecture on diversity estimates, and how present day diversity can be shaped by ancient demographic events.”

Lines 171-172. Does this refer to >100 segregating sites or all sites, including monomorphic positions?

This refers to all sites, and has been clarified in the revised text.

Lines 195. A generation time of 6.5 years for a small passerine birds seems very high. Cf. generation time estimates in other bird species.

We trust our estimate. For example, the oldest birds in the Raso long-term study were ringed as adults in 2002 and caught again twelve years later. We think this is an interesting characteristic of this species, and it actually contributes to the story because the number of generations since the colonisation of Cape Verde by humans is lower than it would be if generation times were short. We have mentioned this in the discussion.

Line 247. The observed heterozygosity is not that low compared to some other birds.

We have removed the absolute statement and just make a relative one:

“we observe strong elevations in the density of heterozygous genotypes across large portions of chromosomes 3, 4a and 5 in all female Raso larks and not in males”

Figure 3 legend. Were the three mentioned skylark samples excluded from all analyses in the paper?

This is a very good point. We have dropped these samples from the entire dataset and have redone all analyses and re-written the manuscript accordingly.

Hans Ellegren

Reviewer(s)' Comments to Author:

Referee: 1

Comments to the Author(s)

The manuscript “Neo-sex chromosomes and demography shape genetic diversity in the Critically Endangered Raso Lark” by Dierickx et al. aimed at understanding the determinants of genetic diversity in the small island population of Raso lark by specifically looking at the role of neo-sex chromosomes and demography.

The authors detect regions of the genome as neo-sex chromosomes using the higher observed heterozygosity in female individuals. They report genetic diversity for the Raso lark and argue that the observed diversity is higher than what would be expected for an island species of such a small population size. They go on to show that by excluding the neo-sex chromosomes, genetic diversity is halved in the Raso lark and consider neo-sex chromosomes as a source of genetic variation. To further explain the observed level of genetic diversity, they conduct demographic modeling in which they show that demographic history of the species is best explained by incorporating both population contraction and an ancient expansion leading to an increase in rare variants.

The authors produced a draft genome assembly for the Eurasian skylark and RAD-seq reads for 78 individuals of 4 lark species which can be used as a valuable resource for further studies on larks and other avian comparative studies.

However, my main concern regarding this manuscript is that the elevated heterozygosity reported in Raso larks is obtained by calculating heterozygosity in neo-sex chromosomes in females. I think this is not correct since the neo-Z and

neo-W chromosomes are diverged from one another and this means that the divergence between neo-Z and neo-W chromosomes contribute to the heterozygosity, i.e., $dxy = 4N_{eu} + 2u_t$ which leads to an overestimation of the effective population size. Moreover, authors argue that recombination suppression has retained genetic variation while this is a known fact that recombination suppression leads to the reduction in genetic diversity. Since the role of neo-sex chromosomes in maintaining genetic variation is a core part of the paper, I am afraid that this result has been incorrectly obtained by using the divergence between neo-Z and neo-W in females.

The referee is absolutely correct that including the divergence between gametologues in genetic diversity calculations to understand effective population size would be flawed. That is a key point in our paper, and we clarify this in the revised version by first addressing the question of neo-sex chromosomes first. In the section on diversity, we now report the correct π (after the regions of suppressed recombination have been excluded). However, we still include the (incorrect) whole-genome π , in order to make the contrast and show how we would have been way off had we not accounted for the neo-sex chromosomes, as this is an easy mistake to make in a non-model taxon of unknown genome structure. All of our demographic analyses already excluded these regions in the original version. Please see responses to specific comments below.

I think this manuscript is an interesting work for the Proceedings of the Royal Society B, however, it requires major revisions before it can be considered for publication as detailed in the comments below:

Introduction:

1. Line 55: Reference 12 gives a mixed message about the genetic diversity and N_e of island species. According to reference 12, some island species show reduced genetic diversity and some show no differences compared to mainland species. I think it would be more comprehensive if both aspects are mentioned in the introduction.

We have corrected this as follows:

“Previous studies have found that some, but not all, island species show reduced genetic diversity and increased inbreeding compared to their mainland counterparts [11,12]. However, this pattern may be driven more by the bottleneck associated with colonisation rather than long-term reduction in N_e [12].”

2. Line 70: A reference is required at the end of the sentence: “Previous work suggests that a change to genome architecture might buffer Raso larks from genetic diversity loss.”

Thank you, this has been added.

3. Lines 71: I think a schematic figure of the *Alauda* neo-sex chromosomes in

the introduction would help the reader to follow the rest of the paper more easily. It would be helpful if the figure shows the recombining and non-recombining part of the neo-sex chromosomes together with the segment used for measuring genetic diversity in females.

We still lack a clear understanding of the structure of *Alauda* sex chromosomes. Apart from linkage mapping showing sex linkage of chromosome 4a in a different genus, and a microsatellite showing sex linkage in the Raso lark, the other components of the neo sex chromosomes were not known before this study. We acknowledge that they have been discovered in parallel by Sigeman et al. (2019, BioRxiv), but that paper also does not reveal fusions etc. We have therefore in fact decided not to make any claims about fusion events in our revised version, and instead only report the regions of suppressed recombination in females.

4. Line 71 to 74: It is stated that “Cessation of recombination on neo-sex chromosomes could represent a source of heterozygosity, because females (the heterogametic sex) could retain distinct alleles at homologous loci on their neo-Z and neo-W chromosomes.” I am confused about this statement. By homologous loci on neo-Z and neo-W chromosomes, are the authors referring to the so-called gametologous genes (homologous genes on the non-recombining sex chromosomes that are not yet degenerated from the W)? First, cessation of recombination leads to a decreased effective population size and therefore, reduction in heterozygosity. Second, due to the lack of recombination, the neo-Z and neo-W sequences have diverged from one another. All females are heterozygous not because two alleles are segregating in the population but because all W sequences are fixed for one allele and all Z sequences for the other allele. This is the result of divergence of two sequences since the cessation of recombination prior to the split of the species studied. It is not therefore correct to measure heterozygosity for females using neo-Z and neo-W sequences since the Tajam’s pi should tell us about the amount of genetic diversity since the most recent common ancestor of the species of interest. Please see for example the analyses in Bachtrog and Charlesworth 2000 regarding neo-sex chromosomes.

We agree in general the reviewer. We think there are two issues here. The first is terminology, and we admit that we should be referring to divergence between gametologues rather than heterozygosity. We have now used this correct term throughout. We also agree that it is not appropriate to consider such variation as contributing to population level diversity in terms of effective population size. In our initial version, because we were unaware of the large extent of the neo-sex chromosomes, we initially computed diversity incorrectly and only later discovered that it was inflated by the neo-sex chromosomes. We had written the paper in this ‘story’ form. We now correctly address the neo-sex chromosomes up front. We do not refer to ‘heterozygosity’, but we do still use our plots of ‘density of heterozygous genotypes’ to reveal the locations of the regions of suppressed recombination in females.

However, we don't believe that the divergence between the gametologues is disqualified as contributing to functional genetic diversity in the broader sense. Although some of this divergence will have accumulated since recombination suppression, some of it reflects pre-existing allelic variation that has effectively been trapped and prevented from loss due to drift. This is analogous to the retention of heterozygosity in functionally asexual hybrid plants like evening primrose. The fact that recombination is suppressed does not automatically mean that the alleles on the neo-W are non-functional. We agree that the neo-W will eventually degenerate, but this is evidently not yet the case: it shares nearly 99% identity with the neo-Z in the regions of suppressed recombination, and therefore likely still contains functional ancestral alleles that will be absent in males.

While diversity among copies of the neo-W may indeed be low due to its reduced N_e , the overall allelic variation retained across both the neo-Z and the neo-W in homologous regions is higher in females. Once these gametologues have diverge so far as to no longer be considered homologous, we agree that it would not make sense to consider this a mechanism to retain genetic variation. However, this is evidently not nearly the case at this stage.

We have carefully reworded our manuscript to more clearly bring these points across. For example:

Results:

“Although the regions of suppressed recombination represent 12% of the genome, estimated genetic diversity is nearly doubled (0.0019) in our dataset of 15 females and 11 males when these regions are not excluded (Table 1). Although the neo-Z and neo-W chromosomes are gametologues, making it arguably incorrect to consider them as contributing to heterozygosity in the conventional sense, their retention of fairly similar sequences, potentially reflecting ancestral standing variation that pre-dates recombination suppression, means that they might still contribute to functional allelic variation and therefore fitness in female Raso larks, as hypothesised previously [17].”

Discussion:

“Nonetheless suppressed recombination might inadvertently result in the maintenance of pre-existing allelic variation between the gametologues that could theoretically contribute to functional diversity and therefore fitness in females [17]. This is analogous to the retention of heterozygosity in functionally asexual hybrid plants [47], except that here it applies only to part of the genome, and only to females. Future work should address whether homologous genes on both gametologues are still expressed and contribute to female fitness.”

Methods:

5. Line 95: It is very good that all scripts are provided in the github page. A master script is however necessary which explains the order scripts should be run. For example, if a researcher downloads the raw data of this study from ENA and wants to reproduce the study, that master script should provide the steps in the pipeline.

The repository has been overhauled, including a comprehensive readme providing all analysis steps.

6. I suggest a reordering of the paragraphs in the Methods section as follows:

- Sample collection
- Whole genome sequencing and assembly
- RAD library preparation
- Sequence processing and alignment
- Pseudo-chromosomal assembly
- Relatedness
- “Allele frequency spectra and genetic diversity” merged with “Proportion of heterozygous sites across the genome” into one subheading
- Demographic inference

We have roughly followed this proposal, except that we have not merged “*Proportion of heterozygous sites across the genome*” with “*Allele frequency spectra and genetic diversity*”, as the former is now included exclusively as a means to study the neo-sex chromosomes, and therefore has been moved to immediately after the assembly section, and renamed “*Identification of sex-linked regions based on heterozygous genotypes*”

Results:

7. Since the sequence of neo-sex chromosomes of several species is available here, it is possible to get an estimate of the timing of recombination suppression by calculating d_s . The authors could add this analysis to their manuscript to improve the inference of the timing of recombination suppression events.

Because we don't have distinct assemblies of the neo-Z and neo-W, our best estimate for divergence between them is the density of heterozygous genotypes in females. Dating the suppression of recombination based on the divergence between the gametologues is challenging and imprecise, because we believe that a large proportion of the existing divergence reflects standing genetic variation that predated the suppressed recombination. We feel that this would be best calculated by producing separate genome assemblies for the neo sex chromosomes and performing a detailed inference of ancestral diversity. We therefore think that this is beyond the scope of the present paper, which is primarily focused on the genetic diversity of the Raso lark.

8. Line 217: What is the percentage of mapped reads from Raso lark to the Eurasian Skylark assembly?

We have now added this information for each individual to Supplementary Table S2 and the main text. Average mapping rates were 88% for skylarks and oriental larks and 89% for Raso larks. Only the outgroup crested larks showed a reduction in mapping rate down to 82% on average.

9. Line 223: Please report Tajima's pi and Watterson theta for each case in males and separately for autosomes and neo-Z chromosomes.

We have added pi and Watterson's theta using only regions of normal recombination and for the whole genome using males only to Table S3. We have not included pi for the regions of suppressed recombination as these clearly vary in their extent among populations, making any comparison fairly meaningless.

10. Line 225: Genetic diversity in the Eurasian skylark from the Netherlands is 0.0097. This tells us about 1 heterozygous SNP in almost every 100 base pairs. This is really a high level of genetic diversity. What is the range of genetic diversity across other studied passerines?

We have added values from a few other passerines for comparison. This value is indeed high, but not exceptional as for example the zebrafinch has similar diversity. However there are risks in reading too-much into these comparisons, especially if different enzymes were used in generating RAD-seq libraries. We therefore focus the paper more on the relative difference between skylark and Raso lark.

How does this high level of genetic diversity might have affected assembly construction?

There are many successful assemblies of more diverse organisms such as insects, including work of some of the authors, so we do not think this is of concern. It is important to ensure that the assembly does not contain duplicated scaffolds that represent distinct haplotypes of the same region. This was confirmed by a low duplicate rate of 1.5% for BUSCO genes.

Did the high level of heterozygosity provide some challenges for the mapping of reads from Raso larks to the Eurasian Skylark assembly?

As noted above, in insects such as Heliconius butterflies, 2%-3% diversity/divergence results in minimal problems for read mapping. Indeed we see no difference in mapping rates among the three Alauda species (now in Table S2). We do see a small drop-off for the outgroup crested lark, but this will not affect our results in a qualitative way as all analyses were performed for each species separately.

11. Line 226: Equation $\theta = 4N_e\mu$ is used to obtain N_e from genetic diversity. The genetic diversity used is 0.0018 which contains the heterozygosity obtained from neo-sex chromosomes in females. I think this might not be correct since the

N_e obtained from $\theta = 4N_e u$ is the time to the most recent common ancestor of the sample. The diversity obtained from the neo-sex chromosomes in females is actually $d_{xy} = 4N_e u + 2ut$. This means that heterozygosity is overestimated by taking into account divergence between sequences leading to an overestimation of N_e . I think higher female heterozygosity must only be used as a method to detect suppressed recombination. Then to calculate heterozygosity in Raso lark, neo-sex chromosomes must be masked and only diversity of autosomes should be used. If sex chromosomes are to be used, then diversity should be calculated for the Z chromosome and W chromosome separately and in calculating N_e , their relative numbers for every male and female should be taken into account. For example, in a standard case, there are 3 Z and 1 W for every 4 autosomes for a male and a female in the population.

We completely agree with everything that the reviewer has said. Our initial calculations were estimates of diversity made before we knew that the extensive sex chromosomes existed and were contributing to this elevated π number. Our original manuscript then goes on to recalculate π correctly as described by the referee, after the existence of the neo-sex chromosomes is shown. We acknowledge that this was perhaps a confusing way to present the paper, as all three referees had the same concern. We have now re-organised it to address the existence of the neo-sex chromosomes first, and we then correctly present π computed from autosomes only. We do still include the 'whole genome' π for comparison, as we hope to indicate to readers that a very wrong impression could be created by not correctly accounting for the large neo-sex chromosomes, and this is an easy mistake to make in non-model taxa where there are no pre-existing genomic resources. In our case, there was no reason to suspect at the outset that chromosomes 3 and 5 were not normal autosomes.

12. Line 245: The heterozygosity measured between neo-Z and neo-W chromosomes in females cannot be compared with heterozygosity between two neo-Z in males. The reason is that the heterozygosity obtained in females is actually divergence between the Z and W sequences. Once recombination is suppressed, the Z and W chromosomes start to differentiate from one another. This is similar to two subpopulations between which migration stops. To calculate genetic diversity, one should use only the Z sequences or the W sequences. Therefore, while elevated heterozygosity in females can be used as a technical way to find regions with suppressed recombination, it does not mean that we have maintenance of diversity due to recombination suppression.

We agree with this statement, and it was our intention in plotting "heterozygosity" (now called "density of heterozygous sites" to explore the evidence for diverged gametologues. We acknowledge that our original version was misleading in implying that this value represents effective population size. However, as noted above, if this divergence between the gametologues also reflects pre-existing diversity that pre-dates the suppression of recombination, then conceptually it is not especially different from heterozygosity in the conventional sense. Given that the gametologues are ~1% diverged, it is likely that distinct functional alleles could exist on both, and these

could theoretically contribute to *functional* genetic variation. A similar argument applies to recently duplicated paralogs. Our revised version states:

“As females carry a single copy of the Z, they should have no heterozygous sites on this chromosome (with the exception of repetitive elements at which mis-mapping can occur, and possibly pseudo-autosomal regions where the W and Z retain homology). The same should be true for neo-sex chromosomes, unless recombination suppression is recent or incomplete, such that the gametologues retain a high level of homology. This would allow RAD-seq reads from the neo-W to map to scaffolds representing the neo-Z (there should be no neo-W scaffolds in our male reference genome). This could potentially lead to an elevation (rather than a reduction) in the density of heterozygous genotype calls in females. Indeed, we observe strong elevations in the density of heterozygous genotypes across large portions of chromosomes 3, 4a and 5 in all female Raso larks and not in males (Figure 2A). This is consistent with the formation of neo-sex chromosomes and subsequent recombination suppression involving these three autosomes. The gametologues have evidently not diverged to the point that sequence homology has been eliminated. Indeed they retain around 99% similarity (1.15 differences per 100 bp on average)”

and

“To study genetic diversity and demography, we used only the genomic scaffolds that show no evidence of suppressed recombination (Figure S3)”

and

“We also considered how different the estimated genetic diversity of Raso larks would be if we did not account for the enlarged neo-sex chromosomes (as may easily occur using a RAD-seq approach without a reference assembly). Although the regions of suppressed recombination represent 12% of the genome, estimated genetic diversity is nearly doubled (0.0019) in our dataset of 15 females and 11 males when these regions are not excluded (Table 1). Although the neo-Z and neo-W chromosomes are gametologues, making it arguably incorrect to consider them as contributing to heterozygosity in the conventional sense, their retention of fairly similar sequences, potentially reflecting ancestral standing variation that pre-dates recombination suppression, means that they might still contribute to functional allelic variation and therefore fitness in female Raso larks, as hypothesised previously [17].”

13. Lines 249 to 258: From “Previously, only party of ...” are not the results of this paper or are the discussion regarding the result, I suggest to move this part to the discussion.

This statement has been changed but not entirely removed as we now describe a search for evidence of fusions based on scaffold alignments to the zebra finch.

14. Lines 275 to 279: From “A similar analysis ...” to “... that excludes chromosomes 5 [38]” are not results of this paper and I suggest to move this section to the discussion.

This previous result is important for our inference of the placement of events on the phylogeny, so we have kept a reduced version of this statement:

“Using our results in combination with those from a recent study of two additional outgroup species, the bearded reedling *Panurus biarmicus* and the horned lark *Eremophila alpestris* [38], we are able to partially reconstruct the stepwise progression of recombination suppression (Figure 2B)”

We think this not unlike phylogenetic studies that use previous work to calibrate their trees, so we think there is a precedent for including this sort of thing in the results section.

15. Line 303: This π without the neo-sex chromosomes is the correct π and this should be reported as the heterozygosity in Raso Larks.

This is now reported as the correct π , as noted above.

16. Line 304: “Recombination suppression across the neo-sex chromosomes therefore does indeed contribute to the maintenance of genetic variation in the Raso lark.”. This sentence is not correct. Recombination suppression has led to a decrease in genetic variation, as mentioned before, one cannot use divergence between two sequences to calculate heterozygosity. I suggest to remove this sentence.

As noted above, we don't entirely agree, as pre-existing variation that pre-dates the recent suppression of recombination could theoretically be preserved. Our point is not to say that θ in the sense of $4N\mu$ is higher due to the recombination suppression (and we agree that diversity among copies of the neo-W will be low due to its reduced N_e), but that it is possible that there is functional allelic variation present in females that has not been lost due to drift as a result of the recombination suppression. We have revised this entire section, as described above.

17. Line 312: It is stated that diversity difference between the Raso lark and the Eurasian skylark “is far greater than expected given the difference in current population sizes between the two species”. This is not surprising because it is the N_e that matters not the census population size. For example, census population size of humans is about 8 billion, that of *Pan tryglodetes* is in the order of tens of

thousands only, yet genetic diversity in chimps is higher than that of humans due to their higher N_e . What is “expected” as a relationship between the census size and genetic diversity?

We have clarified the point we were trying to make, which is exactly the point that the referee makes:

“Although autosomal genetic diversity in Raso larks is lower than that in related species, it remains far greater than would predicted under long-term persistence at its small census size of typically under 1000.”

18. Line 391: I suggest that the Relatedness section to be moved to the beginning of results as it is more of a quality check and it is important to be presented before the estimates of diversity.

Done

Discussion:

19. Line 413: “... this difference is smaller than expected.”. What is the expected difference?

This has been reworded:

“this difference is much smaller than the difference in census population sizes.”

20. Line 436: “... 1-1.5% divergence”. This is a result not reported in the result section. Please add to both Methods and Results the text regarding the calculation of divergence between the neo-Z and neo-W chromosomes.

We now report this in the results section:

“The gametologues [...] retain around 99% similarity (1.15 differences per 100 bp on average)”

21. Line 451: “... has been favored by selection following the loss of genetic diversity.” Please clarify this sentence.

This sentence has been removed in the revised version.

Line 471: “... level of divergence between the neo-Z and neo-W is similar in all three Aluade species.”. This is a result not presented in the result section. If known from previous work, please cite the reference.

This result is now stated in the results section:

“Within the regions of suppressed recombination, the density of heterozygous sites is similar to that in Raso lark females (1.09%)”

Referee: 2

Comments to the Author(s)

Dear Editor:

Dierickx et al. profiled the genetic diversity of different lark population, particularly the endangered Raso Lark concentrated at two islands of Africa, regarding their chromosome-wide polymorphism level. They found signatures of neo-sex chromosomes, which are different between different skylark and lark populations. Then they tried different methods to fit the historical change of population size to the unexpectedly high polymorphism level as compared to its reported population size nowadays. Maybe because the population contraction is too recent, the current data cannot find a clear signature for the recent population contraction. I have several comments about the paper:

1. It is a bit confusing to state that neo-sex chromosomes maintain the heterozygosities in Raso lark, as the author mentioned in several places in the paper. As first neo-sex chromosomes probably formed by genetic drift in this case, due to the population size reduction. And generally, the differences between neo-Z and neo-W are expected to be excessive deleterious mutations that are fixed on the neo-W, rather than something that the population would like to maintain.

While it is true that neo-sex chromosomes can eventually accumulate deleterious mutations, in this case there is good evidence that the divergence between the neo-W and neo-Z is low, as they share 99% identity. We did not intend to suggest that the formation of neo-sex chromosomes was selected for due to an effect on diversity, but we do believe that it is possible that some of the differences between neo-W and neo-Z represent standing variation that was present at the time of recombination suppression. Therefore, it is possible that functional allelic variation is preserved on the neo-W and prevented from loss due to drift:

“Although the neo-Z and neo-W chromosomes are gametologues, making it arguably incorrect to consider them as contributing to heterozygosity in the conventional sense, their retention of fairly similar sequences, potentially reflecting ancestral standing variation that pre-dates recombination suppression, means that they might still contribute to functional allelic variation and therefore fitness in female Raso larks, as hypothesised previously [17].”

We note that this idea was put forward in a previous microsatellite-based study by Brooke et al. 2010 entitled “Widespread Translocation from Autosomes to Sex Chromosomes Preserves Genetic Variability in an Endangered Lark”

In the discussion we clarify that we do not suspect that this phenomenon is in any way associated with the formation of the sex chromosomes.

“Nonetheless suppressed recombination might inadvertently result in the maintenance of pre-existing allelic variation between the gametologues that could theoretically contribute to functional diversity and therefore fitness in females [17]. This is analogous to the retention of heterozygosity in functionally asexual hybrid plants [47], except that here it applies only to part of the genome, and only to females. Future work should address whether homologous genes on both gametologues are still expressed and contribute to female fitness.”

The reviewer’s suggestion that the neo-sex chromosomes may have formed during the period of low N_e in the Raso lark’s history is not consistent with the data, because the mainland species share these structural changes and appear to have had consistently large N_e in the recent past.

2. Given the current chromosomal sequences are using zebra finch as the reference (as the author also pointed out), it is very likely that the current fusion order in Figure 4 would be subjected to change. First, it is difficult to imagine that chr5, which is more distant from chrW than chr3, to first suppress recombination. Because one would expect sexual antagonistic selection would first accumulate nearby the W first, so that they are more likely to be inherited in only one sex.

This is correct. We lack sufficient information to determine the fusion order, and in the revised version we do not show fusions or make any claim about their order. There is no straightforward interpretation since the progression seems to involve multiple distinct parts of both chromosomes 3 and 5. We now consider the possibility that these chromosomes have fragmented, but this would require a more contiguous genome assembly.

“It is important to note that the chromosome map used here - based on the zebra finch genome - does not reflect the true karyotype for larks. While previous work indicates that only a fragment of chromosome 4a has become sex-linked [18], we cannot currently say whether chromosomes 3 and 5 have undergone similar fragmentation. One scaffold appeared to bridge between regions of suppressed and normal recombination (Figure S1), but the two parts mapped to different chromosomes (3 and 2), so we concluded that this is most likely a misassembly. We also lack any direct evidence that chromosomes 3, 4a and 5 (or fragments thereof) have fused to the Z chromosome. One scaffold appears to bridge chromosomes 3 and 5, but this too may be a misassembly, as it maps to the centre of the two regions of suppressed recombination (Figure S1).”

Second, there is a dip of heterozygosity in most populations between the first part and second part of chr3 (Figure 3, Raso lark). Is this caused by sequencing/assembly gaps?

Sigeman et al. (2019 BioRxiv) who discovered these neo sex chromosomes in parallel, found the same pattern using a different assembly, sample and sequencing approach. In fact, their paper emphasises the dip even more clearly than ours.

And also, is the female heterozygosity level of the first part of chr3 significantly different from those of the second part? This is important if they actually formed at separate time points.

The relative timing of these events is addressed in the Sigeman et al. paper. We believe that dating the suppression of recombination based on the divergence between the gametologues is challenging and imprecise, because we don't know what proportion of this divergence reflects standing genetic variation that pre-dated the suppressed recombination. We feel that this would be best calculated by producing separate genome assemblies for the neo sex chromosomes and performing a detailed inference of ancestral diversity. We therefore think that this is beyond the scope of the present paper, which is primarily focused on the genetic diversity of the Raso lark.

Another question, does the author have some explanation for the elevated heterozygosity level of male on the Z chromosome in Eurasian skylark? This is too obvious to be ignored.

This is expected as males are diploid for the Z chromosomes. The value is lower than the autosomal value, which is expected due to the lower effective population size of the Z.

3. Regarding the demographic analyses, I would think the most straightforward way is collect data from nearby islands that used to be inhabited by Raso lark? Would this be possible from the museum specimen?

This is a very exciting idea, and if we had further funding for this project, this would be an exciting prospect for the future, but currently this is beyond the scope of the study.

Referee: 3

Comments to the Author(s)

Dierickx et al. used reduced representation (RAD) sequencing to explore genetic diversity and the population history of raso larks. They used the reference from a close relative, the Eurasian skylark, to assemble the sequence data and also included population RAD data from several species of larks. The study system is interesting from a conservation genetics perspective since there has been a

monitoring of the population size for quite some time and is possible to directly contrast the census size with the diversity in the genome. They found much more genetic variation than expected from such a small population and also that substantial parts of the genome were sex-linked.

I think the manuscript in general is well-written and the figures are good. Despite having pretty limited within-population genetic data by today's standard (low coverage RAD sequence data), they manage to perform an impressive amount of population modelling.

I do think that the heterozygosity plots (Figure 3 and some of the supplemental figures) strongly support the existence of neosex chromosomes, which has also been suggested by earlier studies on lark karyotypes and by sequence data in other Sylvoidea species. The exact boundaries of these chromosome regions may be difficult to determine with this data set because coverage is limited and not all of the scaffolds could be aligned to the zebra finch genome. However, it seems convincing that recombination suppression is different between some of the focal species, particularly between the crested and raso lark on chromosome 3.

At times I feel that there is too much emphasis on the adaptive value of the neosex chromosomes in maintaining variation in the raso lark population (see for example lines 69-74, 420-421 and 450-451). To my understanding, this is not the general idea of how sex chromosomes form and are maintained, but it is rather through sexual antagonism (which is also mentioned briefly on lines 453-455). I think the authors should be a little bit more careful about the variation-buffering interpretation of these chromosomal changes.

We agree. We did not intend to suggest that any adaptive benefit of preserved variation was involved in the formation of the neo-sex chromosomes, and have modified the text accordingly. However, we do still mention the potential inadvertent consequence of suppressed recombination because it was raised by a previous study, and our data show that it is plausible.

“Nonetheless suppressed recombination might inadvertently result in the maintenance of pre-existing allelic variation between the gametologues that could theoretically contribute to functional diversity and therefore fitness in females [17]. This is analogous to the retention of heterozygosity in functionally asexual hybrid plants [47], except that here it applies only to part of the genome, and only to females. Future work should address whether homologous genes on both gametologues are still expressed and contribute to female fitness.”

One potential general concern of the study is that they use low coverage genetic data that is mapped to the reference of another species in the same genus. Using a reference genome of related species is indeed better than a de novo assembly of the RAD data and the authors also use several methods that could deal with low coverage sequence data, such as *angsd*. That said, I don't know if all of the methods they use are robust to the combination of very low coverage

and cross-species mapping. One would expect that overall diversity in the Raso lark could be somewhat underestimated due to the divergence from the reference genome, but I'm not sure if any specific skew would be predicted on the allele frequency spectrum and hence on the demographic inference.

We don't think that diversity is underestimated in the Raso lark, since the read mapping rates are high and similar for all species (88% on *avaergae* for skylark and oriental lark and 89% for Raso lark, These rates are now included in Supplementary Table S2). Mapping rates were slightly reduced for the outgroup crested lark, so could theoretically affect diversity estimates there, but that is not a focus of our paper.

Regarding the effect of low coverage - we used ANGSD, which is designed specifically to deal with this potential concern. Indeed skylark diversity is found to be high, so we don't think we have underestimated it. We also looked at diversity considering only sites in the genome with reasonable coverage (at least 5x), and the result is the same.

On top of the low coverage, RAD data also has some additional potential problems with allele dropouts and sequence duplicates originating from the PCR steps. Since this is single-end data and not paired-end data, there seems to be no easy way to identify and remove potential PCR duplicates. It would be good if the authors could speculate how much of a problem, if any, these things could be.

The risk of PCR duplicates is inherent to all single-end RAD studies. It can't however explain our key results of (1) differences between sexes in neo-sex chromosomes, and (2) higher diversity in Raso larks than predicted by their current population size alone. Also, we avoided biases in library preparation toward any given population by doing libraries prep in several batches that were partly randomized in their species composition.

While allele dropouts can indeed affect RAD-seq data, they are also unlikely to affect our main findings. The patterns of heterozygous genotypes across the genome are averaged over 250 kb windows, which had a median of over 4,000 sites genotyped in at least one individual per population at at least 5x coverage. Therefore, species differences in the presence/absence of individual RAD loci cannot explain the broad patterns seen. We have added the following statement to the results:

"These species differences cannot easily be explained by artefacts such as allele dropout, as the trends we describe are at the scale of multiple megabases, and are therefore supported by tens to hundreds of separate RAD loci."

All other analyses were performed at the within-species level, so dropouts that affect particular species are not a problem. While dropouts affecting certain individuals could affect the site frequency

spectra, we found that the observed spectrum for Raso larks was robust to bootstrapping over sites and to dropping out one individual at a time (Supplementary Figure S5).

Related to this, I think it is important to more explicitly state the estimated divergence (average sequence divergence and/or divergence times in Myrs) between the studied species and specify the mapping rates of the reads from the different species.

The biology of how and when the species diverged is the focus of another study. But we do recognize that divergence between the genomes is an important consideration for the mapping of the RAD-seq reads. We have now added read mapping rates for each individual. They are summarised in the paper as follows:

“The average proportion of reads mapped was 88% for Eurasian and Oriental skylarks, 89% for Raso larks and 82% for the outgroup crested larks.”

Table S2 provides the coverage of the mapped data, but no information is provided on how much trimmed sequence data there were for each sample in the beginning (i.e. sequence data that could potentially be mapped).

The total number of reads and % mapped are now added to Table S2.

Figure 2 shows that when including females the neosex chromosomes have a large impact on the nucleotide diversity estimates in the raso lark. It is not clear to me though if any of the population history analyses, which were based on the inferred allele frequency spectrum, were also run on any of the subsets of the data (i.e. all samples excluding the sex-linked scaffolds or only males). Including the sex-linked SNPs and females must change the shape of the allele frequency spectrum quite a bit and could potentially have a large impact on the population modelling.

Apologies for the lack of clarity. In the revised version we make it clear that diversity and demographic analyses are all computed after removing the regions of suppressed recombination.

Minor comments

Figure S2: The heterozygosity plot of skylark populations from Eastern Russia and Mongolia looks much messier than the other skylark populations. Is it because there is more variation in coverage among these samples, lower coverage overall or something related to quality of these libraries?

This is caused by lower coverage overall in some of these individuals, which leads to more noise because there are fewer sites that meet the minimum coverage requirement of 5x. This has been noted in the figure legend.

These samples should also be the ones that are most closely related to the reference genome. To clarify the origin of the samples please add a note in Table S2 if Russian samples are from the western or eastern part.

We have indicated this by adding (W) or (E) in the table.

Line 121: For allpaths, please specify that the option “—haploidfy=TRUE” was used. This is at least what it says in the code provided on github. This is useful information for others who want to assemble their own data.

Done.

Line 145: Is it possible to clarify what a “mapping quality adjustment of 50” mean? The other settings (mapping and base quality) are more generic. Is it a default setting?

This has been clarified:

“A mapping quality adjustment was applied for reads with multiple mismatches (-C 50’), following the author’s recommendation.”

Lines 213-215: It seems to be a decent Illumina assembly, especially considering that the only mate pair library had an insert size of 3 kb. It would also be good to provide the number of scaffolds in the summary. To get a more independent quality measure, it would also be reasonable to use busco to quantify the number of conserved single copy genes, for example using the vertebrate or bird set.

Thank you for the suggestions, we have added the number of scaffolds and BUSCO results to the paper. Our assembly includes 93% of BUSCO bird orthologs.

Line 224: According to figure 3 there were three skylarks from the Netherlands that were removed because they had an excessive heterozygosity or homozygosity. Have they also been removed when calculating the nucleotide diversity for this population?

Thank you for pointing this out. These have now been removed from the entire study.

Line 424: The estimate of 42.2 million years must be associated with some uncertainty. Change to 42 Myrs or about 40 Myrs.

Done

Lines 428-430: A linkage map is strictly not necessary to provide evidence of fusion events between these chromosomes. Some additional long-range scaffolding information (e.g. provided by optical maps or Hi-C) may be enough, at least for some of them. It is worth checking if there are any scaffolds with increased heterozygosity in females that map both to ancestral Z and chromosome 3/4A/5 in the zebra finch genome. It is unlikely, but I’ve seen the

4A-Z breakpoint in Illumina assemblies comparable to this in two other Sylvoidea species.

We agree that other methods could provide chromosome-level data, so we have amended the paper to say “*without a chromosomal assembly.*”

We had already checked in the mummer output for scaffolds that could potentially indicate a fusion between chromosomes 3/4A/5 with the Z, but we did not find one. We now state this in the results:

“While previous work indicates that only a fragment of chromosome 4a has become sex-linked [18], we cannot currently say whether chromosomes 3 and 5 have undergone similar fragmentation. One scaffold appeared to bridge between regions of suppressed and normal recombination (Figure S1), but the two parts mapped to different chromosomes (3 and 2), so we concluded that this is most likely a misassembly. We also lack any direct evidence that chromosomes 3, 4a and 5 (or fragments thereof) have fused to the Z chromosome. One scaffold appears to bridge chromosomes 3 and 5, but this too may be a misassembly, as it maps to the centre of the two regions of suppressed recombination (Figure S1).”

Lines 442-443: Alternatively, at least for chromosome 5, it is possible that not the entire chromosome was joined to the ancestral Z chromosome, as was the case for 4A.

Thank you. See response above.

Lines 444-445: Clarify that recombination suppression has expanded over a larger interval of chromosome 3.

Both our results and those of Sigeman et al. show that in Eurasian skylark and Raso lark the extent of recombination suppression on chromosome 3 is greater than that in Oriental skylark and red-tailed lark. The section describing this has now been entirely re-written.

Lines 483-485: 100 000 years seems to be an unrealistically low estimate for their colonization of the islands. This species is well diverged from its closest relatives. If the suggested scenario would be true, the presumably large source population from which they had spread from must have become extinct after this colonization and it is unclear why it would have. Unless the timing estimate is very misrepresentative (an order of magnitude too low), I think another biologically more plausible explanation must be used for the population bottleneck.

This is a valid point. We have revised the text as:

“One possibility is that this could coincide with their colonisation of Cape Verde, but it is likely that divergence from the Eurasian skylark occurred much earlier. This question is the focus of an ongoing study.”

Line 485: Is this the total area of the islands or the area within them that would be suitable for raso lark. Or is it so that almost all area is suitable?

We have clarified in the text that this is a rough estimate that uses the total area of islands and does not consider habitat suitability. However, we do not think this is a problem, because we took the whole area of Raso as well (despite certain parts not being suitable).

Appendix B

Responses to Reviewers

We are grateful to the editor and the reviewers for their insightful and constructive comments. We are happy that they valued our paper and that they think our previous corrections have greatly improved it. We have revised our manuscript in order to fully address the last few remaining comments. The changes are summarised as follows:

- 1. We have fixed the language following comments from Reviewer 1, on the lines indicated by the reviewer.**
- 2. We have added Table S2 that we had forgotten to attach to the submission.**
- 3. We have even further toned down the idea that the presence of neo-sex chromosomes may inadvertently have contributed to fitness in females, as recommended by Reviewer 3.**

Specific responses to all comments are given in BOLD below, and a “track changes” version of the manuscript is appended.

Referee: 1

Comments to the Author(s).

The authors have put a considerable effort and made major revisions to the manuscript which has greatly improved the work. The main point of including ZW divergence in the overall diversity is now corrected, the methods read very clearly now and the analyses steps and codes are described and provided in the github repository.

Please find below some minor but I think important comments to be addressed:

(Page and line numbers are provided based on the TrackChanges version of the manuscript)

Results:

Page 8 - Line 273: "The gametologues have evidently not diverged to the point that sequence homology has been eliminated."

The term "homology" has been used incorrectly in this sentences. Homology reflects the evolutionary history. Two characters or genes are homologous if they share a common ancestor and if they are homologous, that homology can never be eliminated. In the case of Z and W genes, since they share a common ancestor, they are always homologous regardless of the degree of degeneration. Instead of homology, "sequence identity" or "sequence similarity" can be used.

This is an excellent point, and we have changed the terminology according to the reviewer’s recommendation.

Page 14 - Line 467: "... indicate fairly strong homology ..."

Same comment as above. Homology is a 0 or 1 situation, it's either there or not. You could modify it simply to "... fairly strong sequence identity/similarity ..."

Again we thank the reviewer, and we have modified our sentence accordingly.

Discussion:

Page 15 - Line 516: "... rule out an increase in mutation rate because the level of divergence between the neo-Z and neo-W is similar ..."

This sentence seems contradictory to me to what the authors argue throughout the paper about female heterozygosity. It is mainly argued throughout the paper that the higher female heterozygosity is a result of maintenance of ancestral genetic variability. However, the message in line 516 is that the observed "female heterozygosity" is actually due to divergence of the W sequence from the Z as a result of their lineage-specific mutations after recombination suppression and not the ancestral polymorphism. It would be great if the authors could clarify this a bit further.

We agree with the reviewer, and to address this comment, we re-wrote lines 516-519 as:

"We cannot currently rule out an increase in mutation rate, but there is strong evidence from our demographic modelling that the population recently contracted from a much larger size, making this the most parsimonious explanation."

Supplementary material page 10: I could not find a separate Excel spreadsheet on Sample information (S2) in the attached files.

We apologize for this mistake and have now added the S2 file. Thank you for pointing this out.

Referee: 2

Comments to the Author(s).

Dear Editor:

When reviewing this paper, I found another group has published the very similar story on the same journal on the larks' sex chromosomes (<https://royalsocietypublishing.org/doi/abs/10.1098/rspb.2019.2051>). Although this work has a slightly different focus on the population dynamics, but the major point, that several chromosome segments have probably fused to the ancestral sex chromosomes of birds in larks, is the same as the published paper. While the published paper produced more data with whole genome sequencing, and provided more detailed analyses on the age of fusion, i.e., the evolutionary strata of the lark sex chromosomes, and then compared these fused chromosomes in terms of other vertebrate sex chromosomes. This draws a broader attention than the current paper that I am reviewing. Thus I am afraid I have to recommend rejection of the paper.

We do not think this argument stands up, for several reasons. Primarily, we and the reviewers have been aware of the Sigeman et al. study since our first submission (which was in fact earlier than Sigeman et al.'s submission to this journal), and it is cited throughout our manuscript. Indeed, Reviewer #3 has even commented on the ages of the strata citing Sigeman et al., but has clearly not reached the same conclusion that our paper is not worth publishing. Secondly, our paper's main focus is the population diversity and dynamics of the Raso lark. Describing the neo-sex chromosomes was an unexpected (and unavoidable) part of the study. Moreover, we also included populations and species not included in the Sigeman et al. paper, revealing more about how the extent of recombination suppression varies among - and even within - species. In fact, Sigeman et al. cite our study on this. Finally, regarding the results that do overlap, we believe that there is great value in having complementary studies that corroborate each other with different methodologies. We therefore think it is entirely unfounded to recommend rejection.

Nevertheless, to emphasise the different focus of our paper, we have re-worded the title to mention diversity and demography before the neo-sex chromosomes.

Referee: 3

Comments to the Author(s).

The authors have addressed most of the issues that I and other reviewers raised for the previous version of the manuscript. I think that the methods and the results are clearer now and that the genetic analyses are more appropriate.

They have also modified the text to tone down the hypothesis that the neosex chromosomes may have a selective benefit of maintaining variation in females. It is now emphasized that this mechanism is unlikely to have been an initial selection pressure for the formation of the neosex chromosomes. However, they keep the idea that the presence of neosex chromosomes may inadvertently have contributed to fitness in females through the maintenance of pre-existing allelic variation that predates the suppression of recombination between the gametologues.

I don't think this is a probable scenario. Based on the recent study of Sigeman et al. (2019) (Repeated sex chromosome evolution in vertebrates supported by expanded avian sex chromosomes. Proc. R. Soc. B 286), which some of the authors here are included in, the recombination cessation appears to have occurred a very long time ago, with the most recent event occurring at least at time the time before the skylark and the raso lark diverged (estimated 6 Myrs or more) and several probably occurring more than 15 Myrs ago. With these time scales I would expect that most of the divergence between the Z and W copies has happened after the recombination cessation and that a number of these changes will have a deleterious effect. The 1 % divergence reported here seems like it could be an underestimate of how different the gametologues really are.

On top of that one would also expect to see some variation in divergence between the evolutionary strata associated with the different chromosome fusions, which is also reported in the Sigeman et al. study. Furthermore, there is no direct measurement of how degenerate the W-linked sequences are in this study. Just because there still is homology between the W and Z copies, it doesn't rule out that there hasn't already been deleterious changes.

To me there seems to be no real evidence supporting the variation-buffering hypothesis, but it rather seems to originate from the observation that this is a small endangered population and that pooling reads from the divergent Z and W copies creates a lot of additional "heterozygosity", which presumably would be beneficial for the population. None of these neo-sex chromosomes are even unique to this species and are shared by the abundant skylark.

In general we think the reviewer makes a valid argument. However, we slightly disagree on this final point: (1) the most recent strata are only shared by Raso lark and skylark, so are quite recent; (2) the fact the skylark also has the neo-sex chromosomes does not mean that they may still have some effect on fitness in the Raso lark.

Nevertheless, we do accept that we have no evidence either for or against the potential positive effect of the female "heterozygosity", and it is indeed sensible to be clear about this in the manuscript. We still believe that the idea has to be mentioned in our paper, because it is an existing hypothesis that was formulated in Brooke et al. (2010), and nothing in our results proves that it is entirely implausible. We addressed this according to the referee's recommendation of "toning it down": we have made it clear that this is an existing hypothesis from Brooke et al. (2010), not ours that we claim to support with new evidence. We have removed any mention of this hypothesis from the results, but we return to it in the discussion as follows:

"It was previously hypothesised that - irrespective of the original cause of recombination suppression - the maintenance of pre-existing allelic variation between the gametologues might now contribute to functional diversity and therefore fitness in female Raso larks [17]. This is analogous to the retention of heterozygosity in functionally asexual hybrid plants [47], except that here it applies only to a portion of the genome, and only in females. While a non-recombining W chromosome is expected to eventually degenerate through gene loss and accumulation of deleterious mutations, the retention of nearly 99% sequence identity between the non-recombining portions of neo-W and neo-Z means that we cannot rule out the presence of functional female-specific alleles on the neo-W. Future work should test this hypothesis by investigating whether homologous genes on both gametologues are still expressed and contribute to female fitness."

That said, I want to emphasize that this is their interpretation of their data and that this doesn't disqualify the results. However, I still wonder if it would not be

more appropriate to tone down this hypothesis even more or even removing it, because it doesn't seem reasonable.

We thank the referee for his very balanced and fair assessment and recommendations regarding this point and our paper in general.